# Global variation in plant-beneficial bacteria in soil under pesticide stress

Danyan Qiu[1,2], Yan Wang[2], Nuohan Xu[1,3], Bingfeng Chen[2], Yuke Zhu[2], Zhenyan Zhang [1,3], Qi Zhang [1,3], Tao Lu [2], Huaping Dong[3], Jianxin Shou[1] & Haifeng Qian [1,2] ✉

The presence of plant-beneficial bacteria (PBB) in soil significantly affects crop production. Excessive agrochemical use in intensive agriculture causes substantial soil residue accumulation, compromising soil health, crop quality, and human health. Understanding changes in beneficial bacteria under pesticide pollution is crucial for guiding sustainable agricultural practices and promoting soil health. We analyze metagenomic data from 1919 soil samples to identify 364 PBBs. We find higher PBB diversity in agricultural soils than in non-agricultural soils; however, pesticide pollution negatively affects the abundance of PBB, particularly those with plant growth-promoting traits. Pesticides not only reduce PBB diversity as individual factors, but they also exert synergistic negative effects with other anthropogenic factors, as determined by Hedges'$d$ effect size and 95% confidence intervals, further accelerating the decline in PBB diversity. Increased pesticide risk also leads to a loss of functional gene diversity in PBB about carbon and nitrogen cycling within essential nutrient cycles, and a reduction in specific amino acid and vitamin synthesis. Artificial application of specific amino acids and vitamins could be an effective strategy to restore PBB in high-pesticide-risk soils. This study provides guidance for regulating pesticide use to mitigate their negative effects on soil PBB and suggests potential remedial measures.

Plant–soil–microbe interactions play a critical role in plant growth, development, and health[1,2]. Soil microbes include both plant-beneficial (PBB) and plant-pathogenic bacteria. Diseases caused by soil-borne pathogens severely reduce global primary productivity and may be further exacerbated by climate change[3,4]. Controlling plant diseases is critical for maintaining agricultural productivity and profitability as they reduce global crop yields by 10%–16% annually, resulting in economic losses of approximately $220 billion[4]. Plants have evolved complex mechanisms to mitigate the effects of pathogens on their growth and acclimatization, including reliance on beneficial bacteria in the rhizosphere and bulk soil[5]. Therefore, understanding the

distribution and abundance of PBB in soil is crucial for enhancing crop production and agroecology sustainability.

A pesticide is any substance, plant protection product, or fungicide used to repel, control, or kill organisms considered pests[6]. The term "pesticide" encompasses herbicides, fungicides, insecticides, molluscicides, rodenticides, and others[7]. In intensive agriculture, pesticides are applied extensively and continuously to meet the growing demand for food[8,9]. Although intensified pesticide application helps prevent harmful crop diseases[10], it also increases the exposure of these compounds to the soil[11], air[12], and aquatic[13] environments and increases the risk of subsequent human exposure. Furthermore, pesticide overuse not only

[1]The Institute for Advanced Study, Shaoxing University, Shaoxing, People's Republic of China. [2]College of Environment, Zhejiang University of Technology, Hangzhou, People's Republic of China. [3]College of Chemistry & Chemical Engineering, Shaoxing University, Shaoxing, People's Republic of China. ✉e-mail: hfqian@zjut.edu.cn

alters the composition of microbial communities, but it also negatively affects their functional potential[14], leading to soil degradation and damaging soil ecology[15,16]. Pesticides (e.g., fungicides) may suppress the activity of beneficial microorganisms, such as arbuscular mycorrhizal fungi, disrupting their symbiotic relationships with plants[17,18]. Pesticides also significantly reduce beneficial bacteria associated with soil and plant immunity, enhance human pathogens, and weaken the soil's ecological stability. Physical properties such as the dissociation constant (pKa), molecular weight, and water solubility of pesticides are found to largely determine the ecological effects of pesticides[19]. On the other hand, increasing pesticide diversity drives the microbial community toward a composition dominated by pesticide-degrading or resistant "opportunists" and "specialists," thus leading to greater network complexity, functional gene restructuring, and even accelerated soil nutrient loss[20]. The functional loss of nutrient-cycling genes can disrupt key ecosystem services by reducing soil fertility, plant nutrient availability, and microbial resilience. Over time, these changes may affect carbon sequestration, greenhouse gas emissions, and overall ecosystem stability. Beyond the active ingredients, pesticide residues may exert prolonged and potentially subtle effects on soil microbiomes. Research shows that pesticide residues are almost exclusively positively associated with the relative abundances of 113 bacterial and 130 fungal taxa, many of which are known pesticide degraders[21]. Therefore, it is critical to elucidate the abundance dynamics and functional responses of beneficial soil bacteria to pesticide stress.

Some probiotics isolated from soil exhibit plant growth-promoting properties and are widely used as alternatives to fertilizers and pesticides in agricultural production[22]. These beneficial bacteria naturally inhabit the rhizosphere or phyllosphere and have been utilized to support plants in various ways. These include suppressing soil-borne diseases[23], regulating plant development, enhancing stress tolerance[24], increasing nutrient availability, and stimulating plant immunity[24]. PBB can be categorized into three groups based on their beneficial properties[25]: (1) biocontrol ability, the ability to mitigate the effects of plant pathogens that hinder plant growth; (2) plant growth-promoting (PGP) activities, to fix nitrogen, solubilize phosphorus and potassium, or produce siderophores and phytohormones; and (3) stress resistance enhancement, to alleviate plant water stress caused by flooding, drought, or elevated salinity. However, the effects of pesticides and their combination with anthropogenic factors on the functions of native PBB in agricultural fields remain unknown. Thus, investigating the diversity and functional changes in PBB in soils contaminated with pesticides could provide guidance for optimizing agrochemical use, thereby reducing the risk of crop diseases.

Based on our previous global assessment showing that pesticide contamination, especially when combined with other anthropogenic factors, greatly reduces soil microbial health[26], we now focus specifically on PBB. In order to understand the effects of pesticide exposure on the global composition and functional potential PBB, we analyze 1919 metagenomic samples from global soils to: (1) investigate the distribution of PBB in agricultural and non-agricultural lands, (2) examine the effects of pesticide risks on PBB traits and composition, (3) assess the synergistic effects of interactions between pesticides and various anthropogenic factors on PBB diversity, (4) examine how the function and soil nutrient cycling of PBB change under pesticide exposure, and (5) identify strategies to restore PBB in the presence of pesticide-induced risks. This study provides valuable insights for improving agricultural practices, safeguarding soil health, and improving crop quality.

## Results and Discussion
### Global taxonomy and distribution of PBB in soils under pesticide stress

The samples used to compile the metagenomic dataset ($n = 1919$) were selected from 88 independent experiments (Supplementary data 1).

These samples were divided into agricultural ($n = 1191$) and non-agricultural ($n = 728$) land according to agricultural practices. We extracted bacteria from these samples to construct a potential PBB database at the genus level following Li et al[25]. (Supplementary data 4). The global geographical distribution of PBB was primarily concentrated in North America, East Asia, and the southern regions of Oceania (Fig. 1a). These areas are characterized by extensive vegetation cover, which promotes the accumulation of PBB. Moreover, the average PBB abundance per sample and α diversity (richness and Shannon indices) were greater in agricultural land than in non-agricultural land (Fig. 1b–e). Compared with non-agricultural soils, agricultural soils often harbor a greater abundance of PBB because of continuous cultivation and management activities, such as fertilizers, irrigation, and crop selection, which provide a richer source of nutrients, habitat, and a healthy competitive environment for beneficial microorganisms. Human activities such as tillage, irrigation, and the application of organic fertilizers introduce multiple sources of microorganisms[27]. Root exudates from crops provide essential nutrients that promote microbial growth, increasing beneficial and pathogenic populations[28]. The application of chemical fertilizers and pesticides alters the composition of microbial communities, favoring certain species[29]. Environmental conditions, such as humidity and pH, further influence microbial dynamics, and create complex interactions within agricultural ecosystems.

To confirm the effects of pesticides on soil microbial communities, we acquired the risk of pesticides in agricultural soil from a global pesticide risk dataset[30] and classified the pesticide risks into three ranks: low, medium, and high. We found that the α diversity (richness and Shannon index) of PBB was highest at low pesticide risk and significantly lower at high pesticide risk (Fig. 2a, b). β-diversity analysis based on Bray–Curtis distances confirmed that the composition of the PBB community differed significantly across pesticide risk levels (Fig. 2c, three-way PERMANOVA analysis, $F (2, 1916) = 6.21$, $R^2 = 0.184$, $p < 0.001$). We performed additional spatial analyses to account for the potential confounding effects of spatial autocorrelation (Supplementary data 5). The Mantel test showed a significant positive correlation between PBB microbial community dissimilarity and geographical distance, indicating the spatial structure of the dataset. After controlling for spatial effects in the partial Mantel test, the correlation between pesticide risk and PBB community composition was found to be no longer significant, thus suggesting that the apparent effect of pesticide exposure on microbial diversity may have been confounded by spatial factors. Furthermore, distance-based redundancy analysis (db-RDA) indicated that both pesticide risk and spatial covariates (PCNM variables) significantly explained the variations in PBB microbial community composition. Notably, pesticide risk remained a significant driver of PBB community composition even after controlling for the spatial structure, highlighting its important role in shaping PBB microbial diversity.

In addition, as pesticide risk increased, the abundance of 14 PBB genera significantly increased, whereas that of 53 PBB genera significantly decreased (two-tailed Welch's t test, FDR-adjusted p < 0.05; detailed statistics in Supplementary data 4), including members of the orders Rhizobiales and Burkholderiales (Fig. 2e, Fig. S1 and Fig. S2). These PBB were primarily associated with PGP traits (Fig. 2d). This result indicated that increased pesticide risk decreased beneficial bacterial functions essential for plant health, predicting a negative effect on soil health due to the overuse of pesticides. These findings suggest that pesticide risk decreased PBB diversity and that varying levels of pesticide risk distinctly shaped soil PBB community structures. As the risk of pesticide use increased, the PBB in soils decreased, highlighting the detrimental effects of pesticide overuse on soil microbial ecosystems. This phenomenon may be one reason why pesticide use diminishes the functional capacity of beneficial plant symbionts[31]. Our results indicated that excessive pesticide use negatively affects the presence of PGP

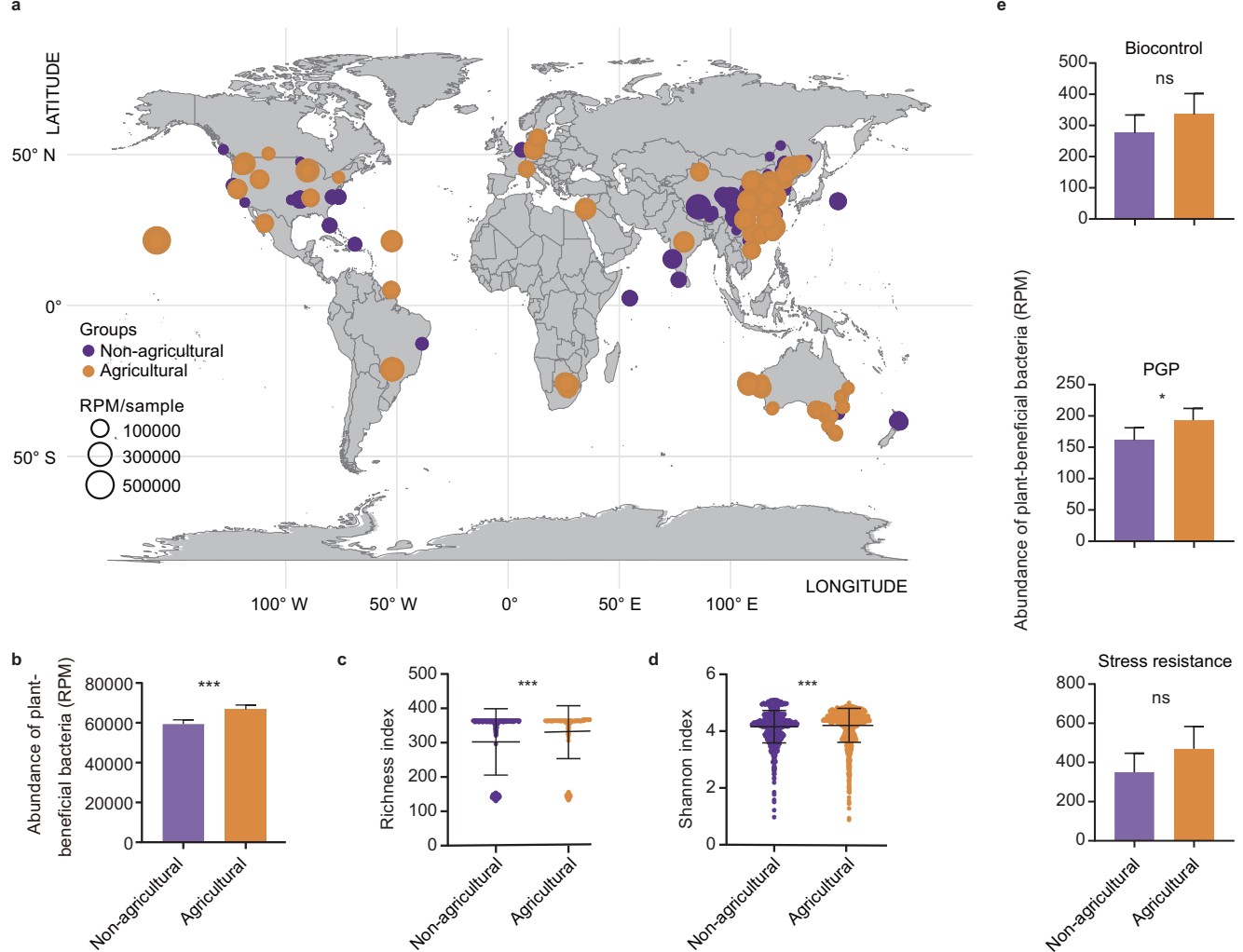

**Fig. 1 | Global distribution of plant-beneficial bacteria (PBB) in agricultural (n = 1191) and non-agricultural (n = 728) lands. **, p < 0.01; ***, p < 0.001. a** Global map of PBB samples. Made with Natural Earth. Free vector and raster map data @ naturalearthdata.com. **b** Abundance of PBB in agricultural (n = 1191) and non-agricultural (n = 728) lands (per sample). The error bars indicate ± standard error of mean (SEM). RPM stands for reads per million., Significance between two groups was evaluated by two-tailed Mann-Whitney U test (p = 1.2e-14). **c** Richness index of PBB in agricultural (n = 1191) and non-agricultural (n = 728) lands. Data are presented as mean ± SD. Significance between two groups was evaluated by two-tailed Mann-Whitney U test (p < 0.001). **d** Shannon index of PBB in agricultural (n = 1191) and non-agricultural (n = 728) lands. Data are presented as mean ± SD. Significance between two groups was evaluated by two-tailed Mann-Whitney U test (p < 0.001). **e** Abundance of PBB with different beneficial traits (PGP, biocontrol, and stress resistance) in agricultural (n = 1191) and non-agricultural (n = 728) lands. The error bars indicate ± SEM. Significance between two groups was evaluated by two-tailed Mann-Whitney U test (pPGP = 9.6e-3). "ns" denotes no statistically significant difference (p > 0.05). Source data are provided as a Source Data file.

bacteria with nitrogen-fixing and phosphorus-solubilizing properties, thereby threatening agricultural productivity. Consequently, we explored the species of PBB enriched or depleted under conditions of pesticide exposure risk and further investigated the effects of excessive pesticide application on PBB survival.

### Effects of interactions between pesticide risk and anthropogenic factors on PBB diversity

For PBB, bacterial diversity decreased as pesticide risk increased (Fig. S3). To capture potential nonlinear and threshold effects on PBB diversity responses, we employed generalized additive models (GAMs). The GAM results revealed distinct nonlinear relationships between the PBB diversity and multiple anthropogenic and environmental predictors (Fig. S4). For instance, pesticide usage exhibited a sharp negative effect beyond a certain threshold, while $CO_2$ emissions showed a unimodal response, thus suggesting that moderate emissions may not suppress diversity, whereas higher levels exert strong negative impacts. Conversely, the GDP showed a consistently positive

association with PBB diversity, potentially reflecting the ecological benefits of economic development. These nonlinear patterns therefore provide further evidence that PBB diversity is not governed by single-factor gradients, but rather shaped by complex, context-dependent interactions among multiple drivers. Variations in anthropogenic factors interact with pesticide stress to jointly affect PBB diversity. Thus, studies on the effect of pesticides alone on PBB diversity are insufficient. An analysis of 12 anthropogenic factors related to climate, human activities, and agricultural practices revealed that the absolute effect size of the interaction between pesticides and anthropogenic factors was significantly greater than that of pesticide action alone (Fig. 3). The combined effects of annual precipitation and pesticides exhibited the most substantial negative effect on bacterial diversity, followed by population density and phosphate fertilizer application. This indicates that the combined impact of precipitation and pesticide risk on PBB diversity loss was greater than the sum of their individual effects, potentially because of mechanisms such as increased pesticide leaching or bioavailability.

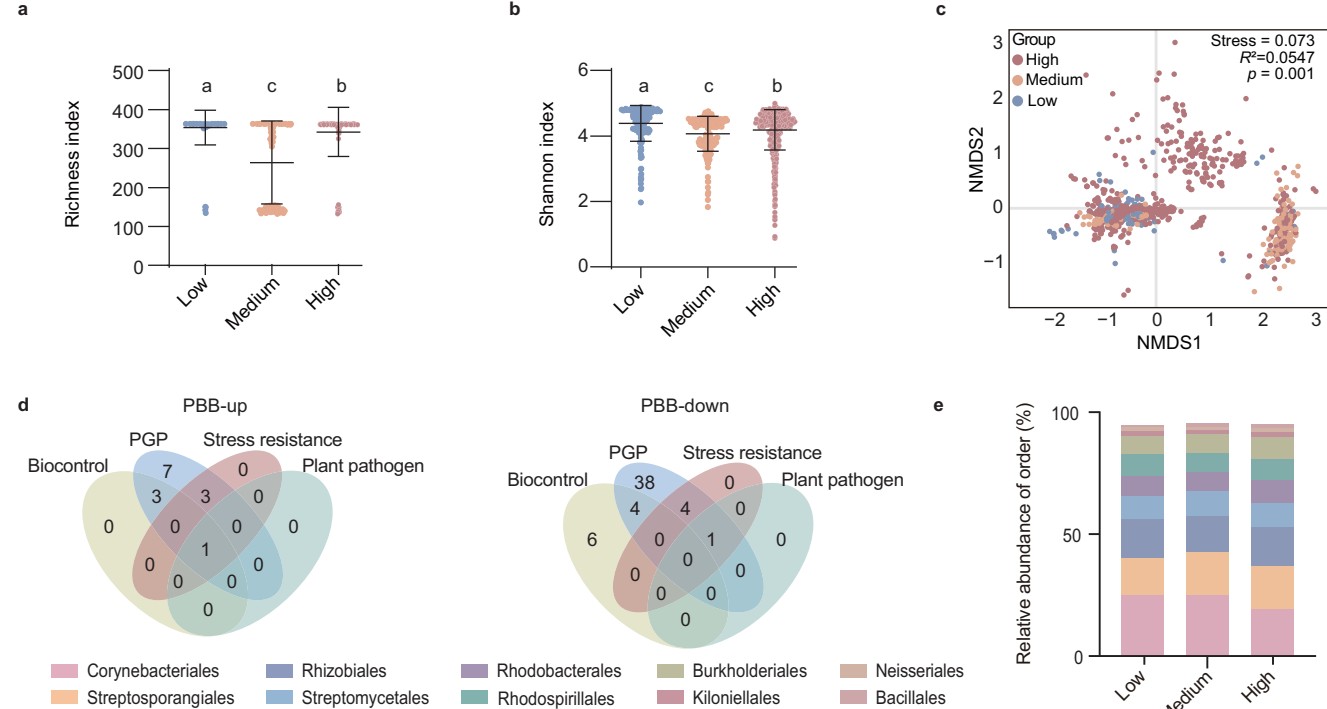

**Fig. 2 | Variation in plant-beneficial bacteria (PBB) on agricultural lands at low (n = 190), medium (n = 214), and high (n = 787) pesticide risk. a** Richness index of PBB at low, medium, and high pesticide risk. Data are presented as mean ± SD. Differences among multiple groups were assessed using the Kruskal–Wallis test followed by Dunn's multiple comparisons test. Significance labels (**a–c**) denote groups that differ significantly from each other. All tests were two-sided, and *p* values were adjusted for multiple comparisons using the Holm–Šidák method. pLow-Medium < 0.0001; pLow-High = 2.3e-4; pMedium-High < 0.0001. **b** Shannon index of plant-beneficial bacteria at low, medium, and high pesticide risk. Data are presented as mean ± SD. Differences among multiple groups were assessed using the Kruskal–Wallis test followed by Dunn's multiple comparisons test. Significance labels (**a–c**) denote groups that differ significantly from each other. All tests were two-sided, and *p*-values were adjusted for multiple comparisons using the

Holm–Šidák method. pLow-Medium < 0.0001; pLow-High = 2.92e-9; pMedium-High = 8.91e-5. **c** Non-metric multidimensional scaling analysis of plant-beneficial bacterial communities on the basis of Bray–Curtis dissimilarities. The effects of pesticide risk on plant-beneficial bacterial communities were assessed via three-way PERMANOVA on the basis of Bray–Curtis dissimilarity. Samples are colored according to pesticide risk. **d** Classification of beneficial traits (PGP, biocontrol, and stress resistance) in PBB that significantly increased or decreased with increasing pesticide risk (*p* < 0.05, two-tailed Welch's t test). PBB-up denotes PBB that increase significantly with the increasing of pesticide risk. PBB-down denotes PBB that decrease significantly with the increasing of pesticide risk. **e** Taxonomic composition of top 10 most abundant bacteria among the 53 significantly enriched PBB at the order level. Source data are provided as a Source Data file.

The interactions among anthropogenic factors suggest that the combined effects of multiple factors amplify the negative effects of pesticides on soil microorganisms, exceeding those of pesticides alone. Pesticide contamination not only diminishes the diversity and functionality of PBB but also increases their sensitivity to other anthropogenic factors, resulting in synergistic effects that further threaten soil health. This synergistic effect suggests that microbial community resilience is influenced by a complex interplay of environmental variables that can exacerbate pesticide-induced diversity loss in agricultural soils. The pronounced negative effect of annual precipitation likely arises from altered soil moisture conditions, which influence pesticide degradation rates and microbial activity, thereby creating harsh conditions for sensitive PBB populations. Population density and phosphate fertilizer application may further strain microbial communities by introducing additional chemical stressors or increasing competition for resources, thereby diminishing PBB diversity. These results underscore the importance of adopting integrated management practices that mitigate the combined stress of pesticides and other anthropogenic factors on soil microbial diversity to maintain long-term soil health and productivity.

### Functional alterations in PBB with pesticide contamination
The function of PBB undergoes significant changes with increasing pesticide risk. The PBB functional gene richness index was significantly lower in the high pesticide risk group than in the low pesticide risk

group (Fig. 4a). Additionally, the Shannon index of functional genes steadily decreased as pesticide risk increased (Fig. 4a), indicating a more pronounced loss in the evenness and diversity of functions across microbial communities. This reduction in microbial functional capacity under high pesticide risk may severely affect soil ecosystem services, such as nutrient cycling[32] and plant–microbe interactions[33,34], affecting the establishment of soil multifunctionality[35] and overall soil health[36].

Comparative analysis of carbon, nitrogen, phosphorus, and sulfur (CNPS)- and PGP-related genes revealed significant functional changes in soil microbial communities with increasing pesticide risk. Compared with those at high pesticide risk, the number of contigs of CNPS- and PGP-related genes was considerably greater than those at low and medium risk (Fig. 4b). The observed decline in key carbon cycle genes, such as K01601_rbcL and K11779_fbiC, along with the nitrogen cycle gene K01429_ureB, is particularly concerning (Fig. 4c). K01601_rbcL encodes ribulose bisphosphate carboxylase/oxygenase, a crucial enzyme for carbon fixation, and its reduction suggests a weakened capacity of the soil to act as a carbon sink. This reduction in carbon fixation impairs the ability of the soil to store carbon, leading to accelerated carbon release, increased atmospheric $CO_2$ levels, and the exacerbation of climate change[37], as reported in previous studies. Moreover, the reduced abundance of nitrogen cycle genes such as K01429_ureB suggests a potential disruption in nitrogen metabolism, that could reduce nitrogen availability to plants and further reduce soil

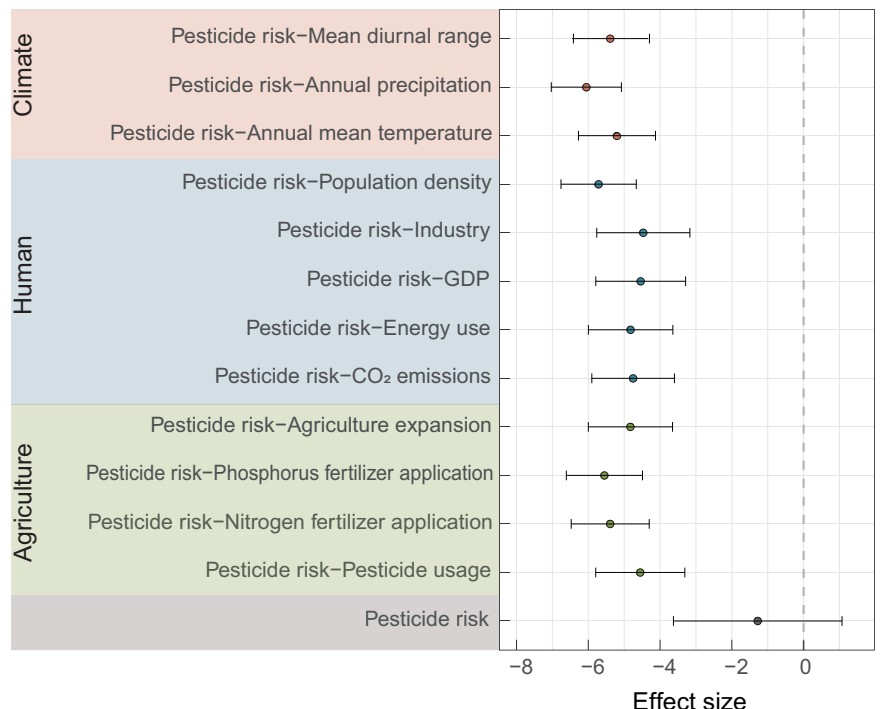

**Fig. 3 | Effects of interactions between pesticide risk and anthropogenic factors on diversity of plant-beneficial bacteria (*n* = 1191).** Interaction between pesticide risk and anthropogenic factors on the basis of the Shannon index of plant-beneficial bacteria. Effect sizes for indicated interaction effects. Data are presented as Hedges'd values ± SD. Source data are provided as a Source Data file.

fertility. Under pesticide-induced stress, the increase in sulfur cycle genes may reflect a shift in microbial strategies to utilize sulfur compounds (e.g., sulfate reduction) as alternative electron acceptors, thereby maintaining energy production and survival. However, this adaptive response is unlikely to offset the broader ecosystem-level disruptions caused by the loss of essential carbon and nitrogen cycle functions. In contrast, the increase in certain carbon (K00239_sdhA, K01183_endoglucanase, K01209_abfA, and K01907_acsA), nitrogen (K00372_nasA, K01428_ureC, K02568_napB, and K15371_gdhA), and sulfur cycle genes (K02048_cysP and K12339_cysM) suggest a microbial response aimed at maintaining functionality under stress, potentially through alternative pathways or compensatory mechanisms (Fig. 4c). However, the loss of critical carbon fixation pathways outweighs these compensatory increases. A decrease in carbon-fixing genes can have severe environmental consequences, such as increased greenhouse gas emissions and decreased soil organic matter[38]. These findings align with those of previous research showing that pesticide exposure reduces microbial diversity and functional gene expression, leading to ecosystem degradation. A reduction in key nutrient cycling processes may compromise soil health, reduce agricultural productivity, and threaten long-term sustainability.

**Functional pathway and growth factor biosynthesis variation in PBB with pesticide contamination**

Owing to changes in PBB abundance and diversity under pesticide exposure, we investigated whether pesticide stress affects functions essential for PBB survival by analyzing the functional pathways and synthesis of growth-essential micronutrients in significantly altered PBB. Eighteen pathways showed significant changes with increasing pesticide risk, and more pathways belonged to PBB-up than to PBB-down (15:3) (Fig. 5a). Conversely, only one pathway, ko00565 (ether lipid metabolism), was enriched in the low pesticide risk group. Among the 18 pathways that showed significant changes, the pathway with the greatest number was the metabolism of cofactors and vitamins (Supplementary data 2). Elevated pesticide risk drives the enrichment of

specific microbial functional pathways, probably due to adaptive strategies that support microbial survival under environmental stress. The observed enrichment of 17 functional pathways under high pesticide risk despite a decline in PBB genetic diversity suggests that pesticide stress may drive the selection of microbial populations with functional redundancies, enabling PBB to maintain critical metabolic processes and enhance their adaptability to adverse environmental conditions. As pesticide concentrations increased, selection pressures favored microbes with resistance and degradation capabilities, enhancing the metabolic and resistance pathways associated with pesticide breakdown and tolerance. Additionally, the toxic effects of pesticides often increase microbial energy and nutrient demands, leading to the activation of pathways related to amino acid synthesis[39], vitamin production[40], and cellular repair processes essential for resilience. The enrichment of pathways involved in redox reactions and detoxification further highlights the microbial defense mechanisms against pesticide-induced oxidative stress. Together, these adaptations enable microbial communities to endure pesticide exposure and partially sustain critical soil ecosystem functions despite heightened environmental challenges.

To identify the growth micronutrients lost due to pesticide exposure in significantly reduced PBB, we manually reconstructed 22 biosynthetic pathways, comprising 625 KO terms, on the basis of the KEGG database (Supplementary data 3). These pathways included 31 essential bacterial growth factors of six types (amino acids, coenzymes, fatty acids, ironophores, nucleotides, coenzymes and vitamins). In the significantly reduced PBB group, the biosynthesis of 19 growth factors was markedly decreased (Fig. 5b), with amino acid biosynthesis (arginine, cysteine, histidine, isoleucine, methionine, proline, threonine, and tyrosine biosynthesis) being the most affected, followed by vitamin biosynthesis (cobalamin, pyridoxal-P, riboflavin, tetrahydrofolate, and thiamine biosynthesis) (Fig. 5c). Vitamins and amino acids are essential for microbial growth and function as coenzymes in metabolic reactions[41]; amino acids are key components of proteins and enzymes[42]. A deficiency of vitamins and amino acids in

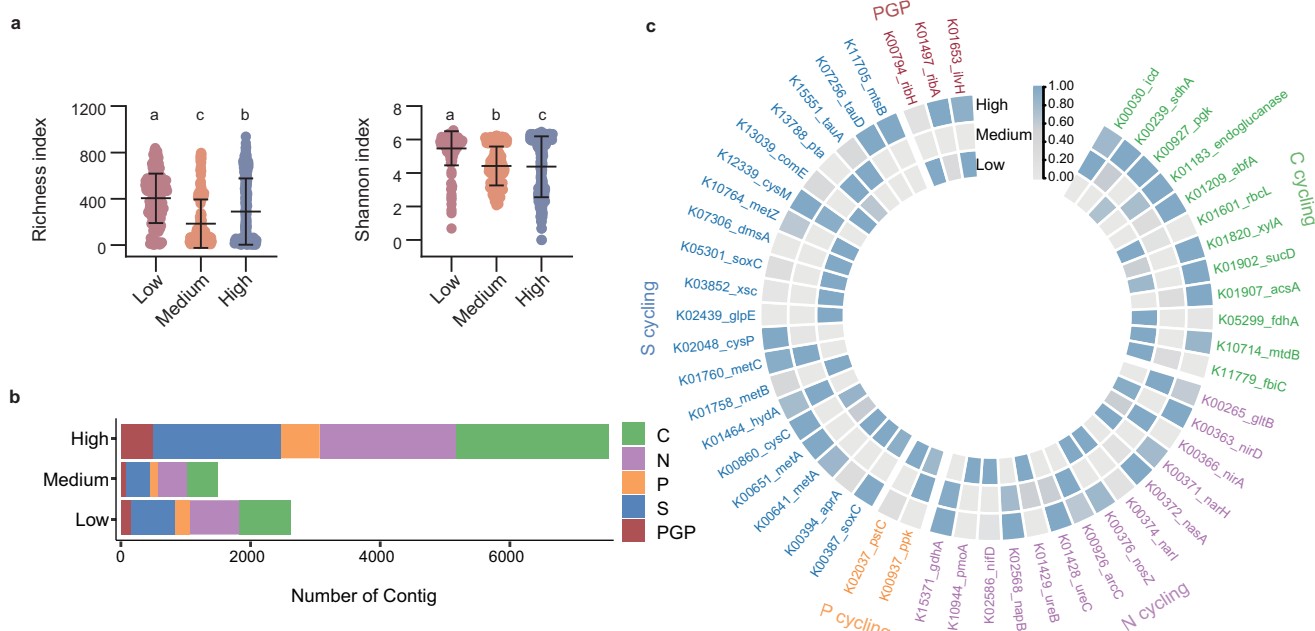

**Fig. 4 | Changes in functional genes in plant-beneficial bacteria at low ($n = 190$), medium ($n = 214$), and high ($n = 787$) pesticide risk. a** Richness and Shannon index of functional genes of differentially altered plant-beneficial bacteria. Data are presented as mean ± SD. Differences among multiple groups were assessed using the Kruskal–Wallis test followed by Dunn's multiple comparisons test. Significance labels (**a–c**) denote groups that differ significantly from each other. All tests were two-sided, and p-values were adjusted for multiple comparisons using the Holm–Šidák method. Richness index: pLow-Medium = 1.24e-10; pLow-High = 1.53e-9; pMedium-High = 1.16e-4. Shannon index: pLow-Medium = 3.99e-10; pLow-High = 2.38e-8; pMedium-High = 2.88e-4. **b** Comparison of contig numbers of plant growth promoting (PGP)-related genes and carbon (C), nitrogen (N), phosphorus (P), and sulfur (S) cycling genes under low, medium, and high pesticide risk. **c** Changes in plant growth promoting (PGP)-related genes and carbon (C), nitrogen (N), phosphorus (P), and sulfur (S) cycling genes under low, medium, and high pesticide risk. Source data are provided as a Source Data file.

soil can impair microbial cell synthesis, reduce metabolic efficiency and energy production, and limit growth rate and population size. Increased pesticide exposure suppresses the synthesis of amino acids and vitamins by PBB, leading to the loss of PBB in the soil. Therefore, the anthropogenic application of these growth factors to pesticide-overused fields could be a strategy for restoring soil PBB abundance and diversity. Although supplementation with amino acids and vitamins represents an innovative strategy for restoring PBB diversity and function, several practical barriers must be considered. These include the economic costs of large-scale supplementation, the feasibility of optimizing application methods and dosages, and also the potential for unintended ecological impacts such as shifts in microbial community composition. Future research should therefore focus on conducting cost-benefit analyses, field trials to refine application protocols, and ecological risk assessments to ensure the sustainable implementation of this approach.

These findings underscore the complex metabolic adjustments that soil microbes undergo in response to pesticide exposure and indicate potential disruptions in critical biosynthetic pathways that sustain microbial life and soil health. Further research is required to explore the ecological consequences of these metabolic changes and their potential feedback effects on soil ecosystem services under various pesticide regimes.

In conclusion, this study generated a comprehensive global profile of 364 PBB species through metagenomic analysis. The findings demonstrate that elevated pesticide contamination in agricultural soils significantly reduces the abundance of PBBs with PGP functions, leading to a decline in critical PBB-mediated soil functions. It is important to note that our functional inferences are based on metagenomic data, which reveal the genetic potential of microbial communities rather than their actual functional activity. While these results indicate possible functional changes under pesticide

stress, future studies incorporating metatranscriptomic analyses are essential to precisely assess gene expression patterns and validate functional impacts at the RNA level. Additionally, our findings suggest that targeted supplementation with specific amino acids and vitamins may serve as a potential strategy for restoring PBB diversity and function in soils exposed to a high pesticide risk, thereby supporting soil health and crop productivity under intensive agricultural practices.

However, the long-term recovery of PBB communities following reduced pesticide use remains unclear. Although some microbial functions may recover over time, persistent pesticide residues or irreversible shifts in microbial community composition can lead to permanent changes in soil ecosystems. Future studies should investigate the resilience and recovery potential of PBB communities under reduced pesticide input scenarios as well as the potential for irreversible microbial shifts caused by prolonged pesticide exposure. These insights are critical for developing sustainable agricultural practices that balance crop productivity and soil health.

## Methods
### Collection and description of samples of soil metagenomes
The keywords "soil AND metagenome," "soil AND microbiome," "farmland AND metagenome," "soil AND metagenomic," "land AND metagenome OR metagenomic," and "soil metagenome OR metagenomic" were searched to obtain the required soil metagenomic data. The databases used included the European Nucleotide Archive (https://ebi.ac.uk/ena/), the National Center for Biotechnology Information (NCBI; https://ncbi.nlm.nih.gov/sra/), and the Metagenome Rapid Annotation via Subsystem Technology (https://www.mg-rast.org/).

Metagenomic data selection was performed based on the following requirements: (1) samples with complete information

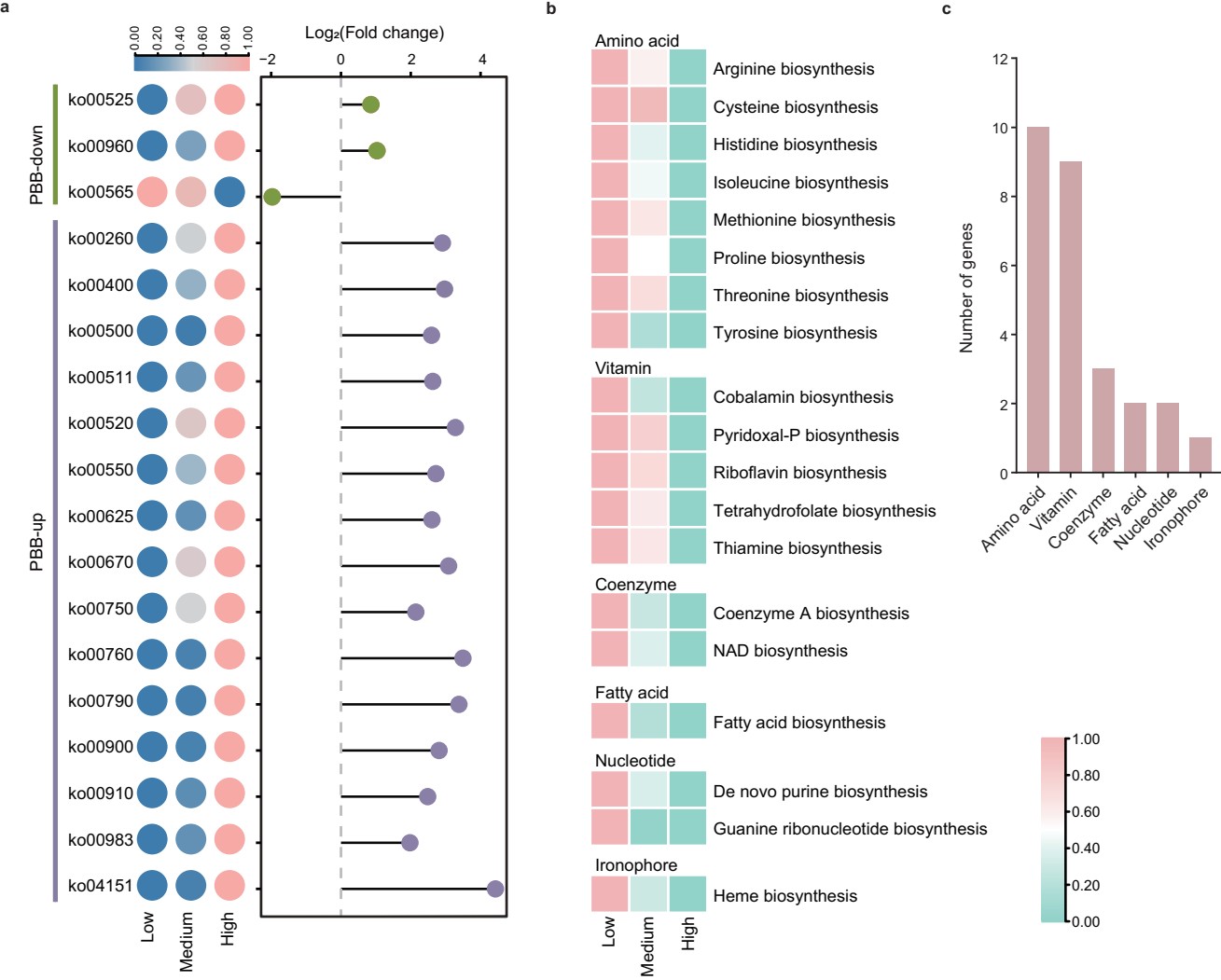

**Fig. 5 | Effects of pesticide risk on functional and growth factor biosynthesis of plant-beneficial bacteria (PBB). a** Functional pathways significantly enriched or reduced with increasing pesticide risk (Kruskal–Wallis test). PBB-up denotes PBB that increases significantly with increasing pesticide risk. PBB-down denotes PBB that decreases significantly with increasing pesticide risk. **b** Significantly reduced growth factor biosynthesis belongs to six biosynthetic categories in significantly reduced PBB with pesticide contamination (Kruskal–Wallis test). **c** Distribution of the number of significantly altered genes across the six biosynthetic categories. Source data are provided as a Source Data file.

(including habitat type, country, latitude and longitude, data source, and biological project number), (2) samples with only pesticide-contaminated exogenous contaminants in the experimental set, and (3) samples closely linked to the plant holobiont, such as root-, stem-, and leaf-colonizing endophytes, were excluded. The aim of these selection requirements was to reduce uncertainty in subsequent analyses by controlling for variables such as sequence quality, sequencing method, experimental protocol, and platform. In total, 1919 samples from 237 global sites were obtained (Supplementary data 1). All raw data are publicly accessible, with file sizes exceeding 2 gigabytes.

**Soil metagenomic quality control, species, and functional gene annotation**

Raw sequencing reads were first processed for quality control by using Trimmomatic (v2.39)[43] to remove adapter and primer sequences, discard reads shorter than 50 bp, and trim low-quality bases (quality score <20). Cleaned reads were then evaluated using FastQC (v0.11.5; https://github.com/s-andrews/FastQC) to ensure that the data quality fulfilled the requirements for subsequent assembly. Quality-controlled reads were assembled de novo into contigs using MEGAHIT[44] (v1.2.8; https://

github.com/voutcn/megahit), applying appropriate k-mer settings to balance accuracy and contig length. Assembly quality was subsequently assessed to ensure the completeness and accuracy of the assembled contigs.

Taxonomic classification of the assembled contigs was performed using Kraken2 (v2.1.2)[45], which assigns each contig to an appropriate taxonomic rank (e.g., phylum, genus, or species). This step provided insights into the microbial community structure within the soil samples.

Functional annotation of the contigs was conducted by mapping to the Kyoto Encyclopedia of Genes and Genomes (KEGG) database, identifying key functional genes and metabolic pathways related to the metabolic capabilities and ecological roles of the microbial community. We further annotated the contigs using a custom database of genes involved in the cycling of carbon, nitrogen, phosphorus, and sulfur[46]. DIAMOND (v2.0.14)[47] was used for sequence alignment (parameters: e-value = 0.001, coverage = 60%, identity = 70%) to assign potential functions to each contig. Based on KEGG, we manually reconstructed 22 biosynthetic pathways comprising 625 KO terms (Supplementary data 3).

In addition to the KEGG-based functional annotation, parallel functional assignments were performed using the COG database (COG2024, NCBI). Protein-coding sequences (CDSs) predicted from the assembled contigs were aligned against the COGorg24.faa reference database using DIAMOND BLASTP (v2.1.3) with default parameters. The best hits were assigned COG identifiers based on their alignment with cog-24.cog.csv, and functional categories and descriptions were annotated using cog-24.def.tab. The relative abundance of each COG ID was calculated per sample, and Kruskal–Wallis tests were performed to detect significantly different COG functions across pesticide risk levels. Significantly enriched pathways were compared using KEGG-based annotations to evaluate functional consistency (Supplementary data 6).

Although we did not benchmark our workflow against other metagenomic analysis platforms such as MG-RAST or QIIME2, all steps were performed using widely adopted and validated tools (e.g., Kraken2 for taxonomy, DIAMOND for functional annotation) that are standard in large-scale soil microbiome studies. The workflow was designed to ensure high-throughput compatibility, scalability, and reproducibility across 1919 metagenomic datasets. All scripts, parameters, and documentation are publicly available on GitHub for transparency and reproducibility.

### Pesticide risk assessment
The pesticide risk assessment in our study was conducted using the PEST-CHEMGRIDSv1 global database[30,48], which provides comprehensive georeferenced data on agricultural pesticide applications. This database integrates multiple data sources, including USGS/PNSP[49] and FAOSTAT[50] pesticide inventories, combined with global gridded datasets of soil physical properties[51], hydroclimatic variables[52], agricultural outputs[53–55], and socioeconomic indicators[56,57]. The risk assessment followed a hierarchical decision-support framework[58], calculating a pesticide risk index as the ratio of the cumulative predicted environmental concentration to the no-effect concentration of each target pesticide. This index quantifies the potential impact of pesticide contamination on soil ecological functions, with values categorized into three separate risk levels: low (index $\leq 1$), medium ($1 <$ index $\leq 3$), and high (index > 3). For each soil microbial sample in our study, the corresponding pesticide risk value was extracted based on the geographical coordinates from this standardized global assessment system, enabling a consistent evaluation of pesticide exposure effects across different regions. The 5 arc-minute resolution (~10 km at the equator) of the database ensured appropriate spatial precision for the analysis of pesticide impacts on soil microbial communities.

### Acquisition of anthropogenic factors and calculation of their effect sizes in combination with pesticides
Pesticide risk calculations were based on PEST-CHEMGRIDSv1 data from 2000 to 2025; therefore, datasets on anthropogenic activities affecting soil ecosystems from the same year were collected from publicly available sources. Twelve anthropogenic stressor datasets were obtained, including climate-, human-, and agriculture-related factors. Climate-related factors, including temperature range, mean annual temperature, and precipitation, were derived from the World-Clim dataset (https://worldclim.org/). Human-related factors such as energy use, population density, and gross domestic product (GDP) were sourced from the World Development Indicators database (https://databank.worldbank.org/source/world-development-indicators). Agriculture-related factors, including agricultural expansion, fertilizer application, and pesticide usage, were obtained from the World At Risk database (https://s3.amazonaws.com/DevByDesign-Web/Maps/DevRisk/index.html#), the Food and Agriculture Organization (FAO; http://fao.org/), and PANGAEA. Total nitrogen and phosphorus fertilizer applications were obtained from

the Harmonized World Soil Database (v1.2) (https://www.fao.org/soils-portal/data-hub/soil-maps-and-databases/harmonized-world-soil-database-v12/en/). The cases for which raster datasets could not provide sufficient detail, regional data corresponding to the sampling sites were used. All variables were standardized to remove scale-related biases.

To capture the potential nonlinear and threshold responses of microbial diversity to environmental gradients, we applied generalized additive modeling (GAMs) using the mgcv and gratie package[59] in R (v4.2.2). GAMs were fitted with a Gaussian error distribution and a log-link function, with microbial Shannon diversity as the response variable, and anthropogenic and environmental predictors as smooth terms. This approach allows the flexible modeling of nonlinear relationships without assuming a priori functional forms. The significance and shape of each smoothed term were assessed using approximate significance tests and also visual inspection of the partial effect plots.

The interaction effect size on diversity between each factor and pesticide risk was calculated using Hedges'd. This standardized mean difference estimate is not affected by sample size[60]. Interaction effect sizes were calculated by comparing the predicted effects with the actual observed effects of individual factors. Calculating interaction effect sizes using absolute values ensured that they were directly comparable, regardless of their direction. Additive effects were defined as interaction effects equal to the sum of their independent effects, synergistic effects as those greater than the sum of their independent effects, and antagonistic or reversal interactions as those less than the sum of their independent effects. The significance of interactions was assessed using 95% confidence intervals calculated around each effect value (from the $t$ distribution). If the confidence interval for an interaction exceeded zero, the interaction was considered additive.

### Statistical analysis and visualization
The diversity (Shannon index) and richness of PBB were calculated using the vegan package[61] in R (v4.2.2). Non-metric multidimensional scaling analysis of variance was performed on the basis of Bray–Curtis dissimilarities in the compositions of PBB and the functional potential (KO level) to determine differences between groups by using the vegan package. The results were visualized using the ggplot2 package[62] in R (v4.2.2). In order to assess whether the observed compositional differences across pesticide risk levels were confounded by spatial autocorrelation, we performed the Mantel test to test the correlation between community dissimilarity (Bray-Curtis distance) and geographical distance. The geographic distance matrix was computed based on the latitude and longitude coordinates of each sample using the geosphere package. We also performed a partial Mantel test to assess the correlation between pesticide risk and microbial community composition, while controlling for spatial distance. Finally, we incorporated the principal coordinates of neighbor matrices (PCNM) into a distance-based redundancy analysis (db-RDA) model. The PCNM variables were derived from the geographic distances between samples using the pcnm() function in the vegan package[61]. The db-RDA model included pesticide risk as the primary explanatory variable and the PCNM spatial variables as covariates. The significance of the model was tested using permutations ($n = 999$). Differential analysis of PBB was performed using STAMP software (v2.1.3). Welch's $t$-test (two-sided) was used to compare differences among treatment groups, with $p$-values adjusted using the Benjamini-Hochberg false discovery rate (FDR) method to control for multiple comparisons. The significance threshold was set at q < 0.05, following FDR correction. All heat maps were drawn using TBtools (v2.119)[63]. The $t$ test and Kruskal–Wallis test were used to identify significant differences via Python (v3.8).

## Reporting summary

Further information on research design is available in the Nature Portfolio Reporting Summary linked to this article.

## Data availability

All metagenomic datasets analyzed in this study ($n = 1919$) were retrieved from public repositories, including National Center for Biotechnology Information (NCBI, https://ncbi.nlm.nih.gov/sra/), European Nucleotide Archive (https://ebi.ac.uk/ena/), and Metagenome Rapid Annotation via Subsystem Technology (https://www.mg-rast.org/). Detailed metadata, including BioProject IDs, ENA Run IDs, collection sites, and environmental parameters, are provided in Supplementary data 1. No new metagenomic data were generated in this study; therefore, no MG-RAST Project ID is applicable. A data file containing the critical supplementary information in this study is provided as Supplementary data 1-6 and is also available in figshare (https://doi.org/10.6084/m9.figshare.30276679). The raw data underlying figures are provided as Source data which can be obtained in public repository (https://doi.org/10.6084/m9.figshare.30276595). Source data are provided with this paper.

## Code availability

The scripts used in this study are available online at https://github.com/QDY742/Global-variation-of-plant-beneficial-bacteria-under-pesticide-stress.git.

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

## Acknowledgements

H.F.Q. was financially supported by the National Key Research and Development Program of China (2022YFD1700401), and the National Natural Science Foundation of China (42307158, 42377107, and 22376187). N.H.X. was financially supported by the Shaoxing Basic Public Welfare Special Project (2024A13001).

## Author contributions

Danyan Qiu: Methodology, Validation, Data curation, Visualization, Formal analysis, Writing-original draft. Yan Wang: Methodology, Data curation, Visualization. Nuohan Xu: Funding acquisition, Methodology. Bingfeng Chen: Methodology. Yuke Zhu: Visualization. Zhenyan Zhang: Methodology, Writing-review & editing. Qi Zhang: Methodology, Funding acquisition, Writing-review & editing. Tao Lu: Resources, Methodology, Writing-review & editing. Huaping Dong: Resources, Writing-review & editing. Jianxin Shou: Resources, Writing-review & editing. Haifeng Qian: Conceptualization, Resources, Methodology, Funding acquisition, Writing-review & editing.

## Competing interests

The authors declare no competing interests.
