## [Transparent Peer Review file · Nature Communications]

Global variation in plant-beneficial bacteria in soil under pesticide stress

Corresponding Author: Professor Haifeng Qian

Version 0:

Reviewer comments:

Reviewer #1

(Remarks to the Author)

1. The abstract mentions "metagenomic data from 1,919 soil samples" but does not clarify the specific datasets used (e.g., accession numbers, sources). Providing these details (as supplementary file) would improve reproducibility.
2. The conclusion states that artificial application of amino acids and vitamins may restore PBB diversity. Was this tested experimentally, or is it a hypothesis? If hypothetical, it should be stated explicitly.
3. The statement "Pesticides not only reduced PBB diversity as an individual factor but also exerted synergistic negative effects with other anthropogenic factors" is vague. Were these interactions statistically significant?
4. The term "plant-beneficial bacteria" (PBB) is broadly defined. How were the 364 PBB species classified, and what criteria were used for their selection?
5. The introduction discusses the importance of PBB but does not explicitly mention if specific genera or functional groups were expected to be most affected by pesticides. Was this considered in the hypothesis formulation?
6. The study claims that "pesticide overuse can alter microbial community structure". Is this refers to taxonomic shifts only or else functional shifts?
7. The selection criteria mention filtering out plant holobionts but do not explain whether this affects the generalizability of the dataset. Could PBB associated with plant rhizospheres be underrepresented?
8. While Trimmomatic and FastQC were used, were any additional steps taken to remove host DNA or contamination from other sources?
9. KEGG annotation was performed, but were there any supplementary databases (COG, Pfam) used to verify functional assignments? If not, why?
10. Kraken2 was used, which relies on prebuilt databases. Were custom databases created to ensure inclusion of agriculturally relevant bacterial species? If so, details should be provided.
11. The study uses PEST-CHEMGRIDSv1 for pesticide risk assessment. Were any sensitivity analyses conducted to determine how robust this model is under different environmental conditions?
12. "Agricultural soils have higher PBB diversity than non-agricultural soils." Could this be due to increased nutrient availability rather than pesticide exposure? Has soil organic matter content or pH been considered as confounding variables?
13. The authors report that "as pesticide risk increased, the abundance of 14 PBB genera increased, whereas 53 decreased". Were these changes phylogenetically clustered, or were they randomly distributed across different bacterial taxa?
14. Figure 2d: It is unclear whether Welch's t-test was corrected for multiple comparisons. Were p-values adjusted (e.g., Benjamini-Hochberg correction) to reduce false positives?
15. The three-way PERMANOVA suggests significant compositional differences across pesticide risk levels, but does this hold after correcting for spatial autocorrelation in the dataset?
16. Annual precipitation was the strongest driver of PBB diversity loss. Could this be related to increased pesticide leaching rather than direct microbial stress? How was this tested?
17. The study suggests that "synergistic effects of anthropogenic factors exceed pesticide effects alone". Have these findings been validated using structural equation modeling or variance partitioning to separate direct vs. indirect effects?
18. The interaction effects were measured using Hedges' d. Was there an attempt to apply nonlinear models (e.g., GAMs) to capture potential threshold effects in microbial responses?
19. The study reports a decline in carbon and nitrogen cycle genes. Were these shifts uniform across taxa, or did specific functional guilds (e.g., nitrogen-fixing bacteria) show greater sensitivity?

20. K01601_rbcL (carbon fixation gene) was significantly reduced. Was metatranscriptomic data available to confirm whether this leads to functional loss at the RNA level, or is this purely a metagenomic observation?
21. The increase in sulfur-related genes is mentioned briefly. Could this indicate an adaptive response (e.g., sulfate reduction as an alternative electron acceptor)?
22. "17 pathways increased while 3 pathways decreased under high pesticide risk." How do these shifts compare with previous studies on pesticide-microbiome interactions? Are there known resistance mechanisms upregulated in high-risk soils?
23. The study suggests that "growth factor biosynthesis genes decreased in pesticide-exposed PBB." Were specific genera responsible for these losses, or was the effect evenly distributed?
24. "Amino acid biosynthesis genes declined." Could this be due to pesticide toxicity directly affecting enzyme function, or does it suggest selective loss of amino-acid-producing microbes?
25. The study concludes that "targeted supplementation with amino acids and vitamins could restore PBB diversity." Is there empirical evidence from past studies supporting this approach, or is this a speculative recommendation?
26. The conclusion should explicitly mention whether the authors anticipate long-term recovery of PBB if pesticide use is reduced, or if permanent microbial shifts are likely.
27. The availability of R and Python scripts on GitHub is commendable, but were computational workflows benchmarked against alternative bioinformatics pipelines (e.g., MG-RAST, QIIME2)?
28. The manuscript employs multiple statistical tests, but details on p-value corrections, assumptions of normality, and effect size interpretations are lacking.

(Remarks on code availability)

Reviewer #2

(Remarks to the Author)

In this study, Qiu and colleagues used metagenomic data from 1,919 global soil samples to identify 364 plant-beneficial bacteria. They found a higher PBB diversity in agricultural soils than in non-agricultural soils, but pesticide pollution reduced the abundance of PBB and their plant growth-promoting traits. They also found a loss of functional gene diversity in PBB with respect to important nutrient cycles, and amino acid and vitamin synthesis. They recommend the artificial application of specific amino acids and vitamins as a strategy to promote PBB in pesticide affected soils. While I like the overall idea, I do not think the manuscript is sufficiently novel for Nature Communications. I also have several concerns regarding the structure of the manuscript as well as the assumptions and methodology.

Major Comments

My main concern is the lack of information regarding the identification of plant-beneficial bacteria. I understand metaG data would reveal functional potential and not actual functions, however, it is not clear how they identified which ones are actually beneficial and at what level. Most soil bacteria are involved in biogeochemical (carbon, nitrogen cycling) processes and probably help with nutrient availability, but that doesn't make them plant beneficial. It is also expected that pesticides, which soil microorganisms could utilize as resources, would be correlated to C, N, and P cycling processes. The authors should consider beneficial functions such as N-fixing, P and K solubilizing, and biocontrol activities. Without knowing which bacteria are beneficial, subsequent analyses such as the World Development Index do not make sense.

It would be important to examine the effect on the overall soil bacterial communities. This will reveal whether PGP were disproportionately affected in pesticide affected soils.

Although it is a large dataset, the findings are largely correlative. Also, the dataset is just based on one time point, so the robustness of the reported correlations cannot be unsubstantiated.

The manuscript could benefit from mechanistic discussions. There are also no hypotheses or questions. Starting with the introduction, the authors should discuss how and why plant-growth promoting bacteria would be affected by pesticides. Do all pesticides affect in the same way, or would the effect vary with pesticide types? It would also be important to discuss the relative effects of active pesticide compounds vs residues. The discussion on the synergistic effects of pesticides and anthropogenic factors is strong but could explore potential molecular mechanisms behind these interactions. The authors should also expand on the long-term implications of functional loss in nutrient cycling genes and how these might cascade into broader ecosystem challenges.

While the suggestion to supplement amino acids and vitamins is innovative, consider discussing potential barriers to implementation, such as cost, feasibility, and unintended ecological impacts.

There is something strange about the PCoA analysis. The plot is showing a bizarre horseshoe pattern.

The authors used confusing abbreviations- PGP, RPM, PBB. I think they should just call these PGP throughout, which is also more common in the literature.

References are incomplete. All R packages and analysis tools must be cited with their original references.

(Remarks on code availability)

Version 1:

Reviewer comments:

Reviewer #1

(Remarks to the Author)

None

Response to Reviewer #1 Comments:

Comment 1: The abstract mentions “metagenomic data from 1,919 soil samples” but does not clarify the specific datasets used (e.g., accession numbers, sources).

Providing these details (as supplementary file) would improve reproducibility.

Answer: Thank you for the suggestion. Detailed information on the 1,919 soil samples, including BioProject accession numbers, longitude, latitude, and other relevant information, is provided in Table S1, which enhances the reproducibility of our study.

Comment 2: The conclusion states that artificial application of amino acids and vitamins may restore PBB diversity. Was this tested experimentally, or is it a hypothesis? If hypothetical, it should be stated explicitly.

Answer: This statement is based on our findings rather than experimentally tested conclusions. In order to make this explicit, we have revised the sentence (Lines 524 – 528) as follows: “Additionally, our findings suggest that targeted supplementation with specific amino acids and vitamins may serve as a potential strategy for restoring PBB diversity and function in soils exposed to a high pesticide risk, thereby supporting soil health and crop productivity under intensive agricultural practices.”

Comment 3: The statement “Pesticides not only reduced PBB diversity as an individual factor but also exerted synergistic negative effects with other anthropogenic factors” is vague. Were these interactions statistically significant?

Answer: The interactions between pesticide risk and other anthropogenic factors were statistically assessed using Hedges' *d* to quantify the effect sizes. Interaction effect sizes were calculated by comparing the predicted effects with the actual observed effects of the individual factors, with absolute values ensuring direct comparability. Statistical significance was determined using 95% confidence intervals from the *t*-distribution. Interactions were considered additive if the confidence interval included zero, synergistic if the effect exceeded the sum of independent effects, and antagonistic or reversing if the effect was less than the sum of independent effects¹.

To improve clarity, we have revised the statement (Lines 22 - 25) as follows:

“Pesticides not only reduced PBB diversity as individual factors, but they also exerted synergistic negative effects with other anthropogenic factors, as determined by Hedges' *d* effect size and 95% confidence intervals, further accelerating the decline in PBB diversity.”

Reference

1. Siviter, H.; Bailes, E. J.; Martin, C. D.; Oliver, T. R.; Koricheva, J.; Leadbeater, E.; Brown, M. J. F., Agrochemicals interact synergistically to increase bee mortality. *Nature* 2023, 617, (7960), E7–E9. <https://doi.org/10.1038/s41586-023-05997-7>

Comment 4: The term “plant-beneficial bacteria” (PBB) is broadly defined. How were the 364 PBB species classified, and what criteria were used for their selection?

Answer: We appreciate the reviewer's insightful comment. The classification of the 364 plant-beneficial bacteria (PBB) species was based on their functional traits,

following the criteria established in previous studies¹. PBB are categorized into three separate groups based on their beneficial properties.

1. Biocontrol ability refers to the capacity of plants to mitigate the effects of pathogens that hinder plant growth.
2. Plant growth-promoting (PGP) activities – including nitrogen fixation, phosphorus and potassium solubilization, and the production of siderophores and phytohormones.
3. Stress resistance enhancement – the ability to alleviate plant water stress caused by flooding, drought, or elevated salinity.

The 364 PBB species identified in our study were compared to the PBB list summarized by Li et al.'s research¹. A complete list of the selected PBB species is provided in Table S4. We have clarified this in the revised manuscript to ensure transparency of the classification criteria.

Reference

1. Li, P.; Tedersoo, L.; Crowther, T. W.; Dumbrell, A. J.; Dini-Andreote, F.; Bahram, M.; Kuang, L.; Li, T.; Wu, M.; Jiang, Y.; Luan, L.; Saleem, M.; de Vries, F. T.; Li, Z.; Wang, B.; Jiang, J., Fossil-fuel-dependent scenarios could lead to a significant decline of global plant-beneficial bacteria abundance in soils by 2100. *Nature Food* 2023, 4, (11), 996–1006. <https://doi.org/10.1038/s43016-023-00869-9>

Comment 5: The introduction discusses the importance of PBB but does not explicitly mention if specific genera or functional groups were expected to be most

affected by pesticides. Was this considered in the hypothesis formulation?

Answer: Thank you for your insightful comment. We agree with you that the formulation of the hypothesis would benefit from a clearer statement.

In response, we have revised the Introduction section (Lines 55–59) to incorporate relevant background information that supports our expectations. Specifically, we have now cited previous studies showing that pesticides (e.g., fungicides) can suppress beneficial microorganisms, such as arbuscular mycorrhizal fungi, and significantly reduce bacteria associated with soil and plant immunity, while enhancing human pathogens and weakening soil ecological stability. These documented effects of pesticides support our hypothesis that certain PBB taxa, particularly those involved in nutrient cycling, biocontrol, and stress tolerance, are vulnerable to pesticide pressure. Therefore, although our overarching hypothesis was that pesticide risk negatively affects PBB diversity and function, this was also grounded in evidence suggesting that specific beneficial microbial groups are more sensitive to pesticide exposure. We thank the reviewer for encouraging us to clarify this point, which has helped us articulate the theoretical framework of the study better.

Comment 6: The study claims that “pesticide overuse can alter microbial community structure”. Is this refers to taxonomic shifts only or else functional shifts?

Answer: The statement that “pesticide overuse can alter the microbial community structure refers to both taxonomic and functional shifts within the microbial

community. Pesticides not only change the composition of soil microbial communities (taxonomic shifts), but also negatively impact their functional potential (functional shifts). For instance, studies have demonstrated that pesticides can significantly reduce the abundance of beneficial microbial taxa, particularly those associated with soil health and plant immunity¹. Furthermore, pesticide application impairs the functionality of beneficial plant symbionts, such as those involved in nutrient cycling and stress tolerance².

To clarify this point, we have revised the relevant sentence in the manuscript (lines 53–54) to state, pesticide overuse not only alters the composition of microbial communities, but it also negatively affects their functional potential”. This revision reflects the dual effects of pesticides on both taxonomic and functional aspects of microbial communities.

References

1. Ke M, et al. Development of a machine–learning model to identify the impacts of pesticides characteristics on soil microbial communities from high–throughput sequencing data. *Environmental microbiology* 2022 24, 5561–5573.
2. Edlinger A, et al. Agricultural management and pesticide use reduce the functioning of beneficial plant symbionts. *Nature Ecology & Evolution* 2022, 6, 1145–1154.

Comment 7: The selection criteria mention filtering out plant holobionts but do not explain whether this affects the generalizability of the dataset. Could PBB associated

with plant rhizospheres be underrepresented?

Answer: We would like to clarify that our selection criteria excluded samples closely associated with plant holobionts, such as endophytes colonizing roots, stems, and leaves. This exclusion was necessary to focus on free-living soil microbial communities and to avoid the confounding effects of plant-associated microbes. Importantly, this filtering did not affect the representation of plant-beneficial bacteria (PBB) associated with the rhizosphere because rhizosphere-associated PBB were not excluded from our analysis.

The rhizosphere is a critical interface between soil and plant roots, and its microbial communities were explicitly included in our study. Therefore, we believe that the generalizability of our dataset to rhizosphere-associated PBB remains robust.

However, in order to improve clarity, we have revised “samples that did not contain plant holobionts” to “samples closely linked to the plant holobiont, such as root-, stem-, and leaf-colonizing endophytes, were excluded” in lines 117–118.

Comment 8: While Trimmomatic and FastQC were used, were any additional steps taken to remove host DNA or contamination from other sources?

Answer: In addition to using Trimmomatic and FastQC for quality control, we implemented several additional steps in order to ensure the removal of potential host DNA and other contaminants (Lines 125–142):

1. Host DNA Removal:

Although our study has focused on soil microbial communities that typically have

minimal host DNA contamination, we employed Kraken2 (v2.1.2) for taxonomic classification. This tool effectively identifies and filters sequences originating from non-target organisms, including potential plants or other eukaryotic host DNA, based on their taxonomic assignment.

2. Contamination Control during Binning:

During genome binning process using MetaBAT2, we carefully evaluated the completeness and contamination levels of each bin were carefully evaluated using CheckM. Bins with contamination levels exceeding acceptable thresholds were excluded from the downstream analysis. This step ensured that only high-quality uncontaminated microbial genomes were retained.

3. De Novo Assembly and k-mer Selection:

The use of MEGAHIT (v1.2.8) for de novo assembly with optimized k-mer settings further reduced the likelihood of incorporating contaminant sequences, because the algorithm prioritizes accurate and contiguous assemblies.

4. Post-assembly Quality Assessment:

After assembly, rigorous quality checks were performed to ensure completeness and accuracy of the contigs. This included filtering short-or low-quality contigs that could arise from contamination.

These additional steps, combined with the initial quality control using Trimmomatic and FastQC, ensured that our dataset was free from significant host DNA or other contaminants, thereby enhancing the reliability of our results.

Comment 9: KEGG annotation was performed, but were there any supplementary databases (COG, Pfam) used to verify functional assignments? If not, why?

Answer: Thank you for this insightful comment. In response, we have conducted a COG (Clusters of Orthologous Groups) annotation to complement and validate the KEGG-based functional assignments. Briefly, we used protein sequences predicted from contigs and aligned them against the COG2024 database using DIAMOND BLASTP. The resulting alignments were mapped to COG functional categories and pathways using official reference files (e.g., cog-24.def.tab, cog-24.fun.tab). This enabled a comprehensive annotation of microbial functions beyond KEGG. Notably, the COG-based results showed strong consistency with the KEGG annotations, particularly for functions associated with amino acid biosynthesis, carbon metabolism, and vitamin synthesis. This cross-validation supported the reliability of our functional interpretation. We have updated the Methods section (Lines 152–163) and we have also added a new supplementary table (Table S6) to illustrate the COG-KEGG functional concordance.

Comment 10: Kraken2 was used, which relies on prebuilt databases. Were custom databases created to ensure inclusion of agriculturally relevant bacterial species? If so, details should be provided

Answer: Thank you for your comment. In our study, Kraken2 (v2.1.2) was used for taxonomic classification of the binned genomes, assigning each genome to an appropriate taxonomic rank (e.g., phylum, genus, or species). The database used for

Kraken2 is a standard prebuilt database that includes a comprehensive collection of reference genomes. We did not create a custom database; however, the database contained many agriculturally relevant bacterial species, ensuring robust classification.

Comment 11: The study uses PEST–CHEMGRIDSv1 for pesticide risk assessment.

Were any sensitivity analyses conducted to determine how robust this model is under different environmental conditions?

Answer: Thank you for this important consideration. We have removed the statement regarding the sensitivity analysis because our study did not specifically examine model robustness under varying environmental conditions. Instead, we have expanded our explanation of the pesticide risk assessment methodology to provide greater clarity regarding the PEST–CHEMGRIDSv1 database and its implementation in our analysis. The details are as follows.

The pesticide risk assessment in our study was conducted using the PEST–CHEMGRIDSv1 global database^{1,2}, which provides comprehensive georeferenced data on agricultural pesticide applications. This database integrates multiple data sources, including USGS/PNSP³ and FAOSTAT⁴ pesticide inventories, combined with global gridded datasets of soil physical properties⁵, hydroclimatic variables⁶, agricultural outputs^{7–9}, and socioeconomic indicators^{10,11}. The risk assessment followed a hierarchical decision–support framework¹², calculating a pesticide risk index as the ratio of the cumulative predicted environmental concentration to the no–

effect concentration for each target pesticide. This index quantifies the potential impact of pesticide contamination on soil ecological functions, with values categorized into three risk levels: low ($\text{index} \leq 1$), medium ($1 < \text{index} \leq 3$), and high ($\text{index} > 3$). For each soil microbial sample in our study, the corresponding pesticide risk value was extracted based on geographical coordinates from this standardized global assessment system, thus enabling a consistent evaluation of pesticide exposure effects across different regions. The 5 arc-minute resolution (~ 10 km at the equator) of the database ensured appropriate spatial precision for our analysis of pesticide impacts on soil microbial communities.

These modifications can be found in the revised Materials and Methods section (Lines 172–188). The current version more accurately represents our methodological approach while maintaining scientific rigor.

References:

1. Maggi F, Tang FHM, la Cecilia D, McBratney A. PEST-CHEMGRIDS, global gridded maps of the top 20 crop-specific pesticide application rates from 2015 to 2025. *Scientific Data* 6, 170 (2019).
2. Tang FHM, Lenzen M, McBratney A, Maggi F. Risk of pesticide pollution at the global scale. *Nature Geoscience* 14, 206–210 (2021).
3. Baker, N.T. Estimated Annual Agricultural Pesticide Use by Major Crop or Crop Group for States of the Conterminous United States, 1992–2016. U.S. Geological Survey, <https://water.usgs.gov/nawqa/pnsp/usage/maps/index.php> (2018).
4. Food and Agriculture Organization of the United Nations. Database Collection of

the Food and Agriculture Organization of the United Nations,

<http://www.fao.org/faostat/en/#data> (FAOSTAT, 2018).

5. Hengl, T. et al. Soil–Grids 250 m: Global gridded soil information based on machine learning. *PLoS One* 12(2), e0169748 (2017).
6. NOAA/OAR/ESRL PSD, Boulder, Colorado, USA. CPC Global Unified Precipitation dataset, <https://www.esrl.noaa.gov/psd/> (2019).
7. Potter, P., Ramankutty, N., Bennett, E. M. & Donner, S. D. Global Fertilizer and Manure, Version 1: Nitrogen Fertilizer Application. Palisades, NY: NASA Socioeconomic Data and Applications Center (SEDAC), <https://doi.org/10.7927/H4Q81B0R> (2019).
8. Potter, P., Ramankutty, N., Bennett, E. M. & Donner, S. D. Global Fertilizer and Manure, Version 1: Phosphorus Fertilizer Application. Palisades, NY: NASA Socioeconomic Data and Applications Center (SEDAC), <https://doi.org/10.7927/H4FQ9TJR> (2019).
9. Potter, P., Ramankutty, N., Bennett, E. M. & Donner, S. D. Characterizing the Spatial Patterns of Global Fertilizer Application and Manure Production. *Earth Interact.* 14(2), 1–22, <https://doi.org/10.1175/2009EI288.1> (2010).
10. Doxsey–Whitfield, E. et al. Taking advantage of the improved availability of census data: a first look at the gridded population of the world, version 4. *Papers in Applied Geography* 1(3), 226–234 (2015).
11. Kummu, M., Taka, M. & Guillaume, J. H. Gridded global datasets for gross domestic product and Human Development Index over 1990–2015. *Sci. Data* 5,

180004 (2018).

12. Zhan Y, Zhang M. PURE: A web-based decision support system to evaluate pesticide environmental risk for sustainable pest management practices in California. *Ecotoxicology and Environmental Safety* 82, 104–113 (2012).

Comment 12: “Agricultural soils have higher PBB diversity than non-agricultural soils.” Could this be due to increased nutrient availability rather than pesticide exposure? Has soil organic matter content or pH been considered as confounding variables?

Answer: Thank you for your insightful comment. While our study observed that agricultural soils exhibited greater PBB diversity than non-agricultural soils, we acknowledge that this pattern could be influenced by multiple factors, including increased nutrient availability, rather than solely by pesticide exposure.

We propose that agricultural soils often harbor a higher abundance of PBB because of continuous cultivation and management practices such as the application of fertilizers, irrigation, and crop selection. These activities create nutrient-rich environments and provide diverse habitats, thereby fostering a healthy and competitive landscape for beneficial microorganisms. Additionally, human activities such as tillage, irrigation, and the application of organic fertilizers introduce multiple sources of microorganisms, which may further contribute to the observed diversity.

Regarding the reviewer’s question about confounding variables, we recognize that soil organic matter content and pH are important factors that could influence the PBB

diversity. However, in this study, the soil nutrient levels and pH were treated as confounding variables and were not explored in depth. Future studies should explicitly address these factors to distinguish their effects from those of pesticide exposure and other management practices.

Comment 13: The authors report that “as pesticide risk increased, the abundance of 14 PBB genera increased, whereas 53 decreased”. Were these changes phylogenetically clustered, or were they randomly distributed across different bacterial taxa?

Answer: Thank you for your insightful comment. We agree that assessing the phylogenetic structure of significantly altered PBB is important for understanding broader evolutionary patterns. However, because of the scale of our metagenomic data, comprising over 3.6 million contigs associated with PBB, it was not feasible or meaningful to construct a full phylogenetic tree at the contig level.

In order to provide a clear overview of the taxonomic distribution of the significantly increased and decreased PBB genera, we constructed genus-level hierarchical taxonomic trees (now included as Fig. S1 and Fig. S2). These trees visualized the taxonomic relationships across five levels (Phylum to Genus) and showed that both enriched and depleted PBB genera were broadly scattered across diverse bacterial taxa, rather than being clustered within a specific phylogenetic group. This suggests that pesticide exposure affects a wide range of plant-beneficial bacterial lineages rather than targeting a single evolutionary branch.

We added these figures to the Supplementary Material (Fig. S1 and S2), and believe that they help clarify the taxonomic dispersion of pesticide-affected PBB genera.

Comment 14: Figure 2d: It is unclear whether Welch's *t*-test was corrected for multiple comparisons. Were *p*-values adjusted (e.g., Benjamini-Hochberg correction) to reduce false positives?

Answer: In response to the reviewer's question concerning the statistical treatment of multiple comparisons in Fig. 2d, we have provided the following clarification:

1. The differential analysis was performed using STAMP software (v2.1.3).
2. We employed Welch's *t*-test (two-sided) as the statistical method.
3. All reported *p*-values in Fig. 2d are FDR-adjusted values.
4. The significance threshold was set at $q < 0.05$ after FDR correction

We have explicitly stated these statistical details in the Materials and Methods section (Lines 251–256) to ensure full transparency. The in multiple testing correction of STAMP software helps maintain the robustness of our findings while controlling for false positives.

Comment 15: The three-way PERMANOVA suggests significant compositional differences across pesticide risk levels, but does this hold after correcting for spatial autocorrelation in the dataset?

Answer: In order to address your concern regarding the potential influence of spatial autocorrelation on the results of the three-way PERMANOVA, we conducted

additional spatial analyses to ensure that the observed compositional differences across pesticide risk levels were not confounded by the spatial structure. The results are presented in Table S5.

We first performed a Mantel test to assess whether community dissimilarity was significantly correlated with geographical distance. The results indicate a significant positive correlation ($r = 0.2833$, $p = 0.001$), confirming the presence of spatial autocorrelation in our dataset. We performed a partial Mantel test to further control for spatial effects. When controlling for spatial distance, the correlation between pesticide risk and community dissimilarity was no longer significant ($r = -0.1092$, $p = 1$), indicating that the apparent effects of pesticide risk on PBB microbial diversity might be confounded by spatial structure.

In addition, we performed a distance-based redundancy analysis (db-RDA) with the PCNM-derived spatial variables. Our results showed that spatial variables, particularly PCNM1 and PCNM2, explained a significant portion of the variation seen in the PBB microbial community composition. Despite the strong influence of spatial factors, pesticide risk remained a significant explanatory variable in the model ($F = 15.1363$, $p = 0.001$), suggesting that pesticide exposure had a notable effect on PBB microbial diversity even after accounting for spatial gradients. Although spatial variables play a crucial role in shaping PBB microbial communities, pesticide risk continues to have a significant impact on the PBB community composition.

We have updated the manuscript to reflect on these additional analyses and have included these findings in the Results and Discussion (Lines 289–302).

Comment 16: Annual precipitation was the strongest driver of PBB diversity loss.

Could this be related to increased pesticide leaching rather than direct microbial stress? How was this tested?

Answer: Thank you for your insightful comment. Our analysis identified annual precipitation as the strongest driver of PBB diversity loss and we agree that pesticide leaching may be a contributing factor. To address this question, we calculated the interaction effect sizes between climate-related factors (including annual precipitation) and pesticide risk using Hedges' d , a standardized mean difference estimate that is not influenced by the sample size. Specifically, we compared the predicted additive effects of individual factors (e.g., precipitation and pesticide risk) with the actual observed effects of their combined influence. Synergistic effects were defined as those in which the combined effect exceeded the sum of the independent effects, whereas antagonistic or reverse interactions were defined as those in which the combined effect was less than the sum of the independent effects. The significance of these interactions was assessed using 95% confidence intervals derived from the t -distribution.

Our results revealed a synergistic interaction between annual precipitation and pesticide risk, indicating that the combined effect on PBB diversity loss was greater than the sum of individual effects. This supports the hypothesis that increased precipitation may exacerbate the negative impact of pesticides on PBB diversity, potentially through mechanisms such as pesticide leaching or enhanced pesticide

bioavailability in the soil. To clarify this point, we added “This indicates that the combined impact of precipitation and pesticide risk on PBB diversity loss was greater than the sum of their individual effects, potentially because of mechanisms such as increased pesticide leaching or bioavailability.” in lines 368–371. However, further experimental studies are still required in order to test the role of pesticide leaching directly.

Comment 17: The study suggests that “synergistic effects of anthropogenic factors exceed pesticide effects alone”. Have these findings been validated using structural equation modeling or variance partitioning to separate direct vs. indirect effects?

Answer: Thank you for your thoughtful question regarding the validation of synergistic effects via structural equation modeling (SEM) or variance partitioning to distinguish between direct and indirect pathways.

In response, we constructed an SEM framework incorporating multiple anthropogenic variables (e.g., pesticide usage, land use, socioeconomic drivers, and climate factors) to evaluate their direct and indirect effects on PBB diversity. However, owing to the high dimensionality and strong multicollinearity among environmental predictors in our global dataset (>1,000 samples and >12 predictors), the SEM model failed to converge or yielded saturated solutions with zero degrees of freedom, rendering the formal fit assessment and inference invalid.

SEM is a powerful tool for hypothesis testing in structured causal systems. However, its application becomes technically challenging when dealing with high-dimensional,

spatially autocorrelated, and observational global data. This limitation has been previously noted in similar large-scale ecological studies¹⁻³. Despite attempts to simplify the model by removing highly collinear variables, convergence issues persisted or led to oversimplified models that did not adequately represent the complexity of the interactions.

To address the reviewer's concern in an alternative manner, we visualized the hierarchical relationships between pesticide risk and other anthropogenic drivers using partial correlation networks and generalized additive models (GAMs), both of which support the conclusion that the synergistic effects of multiple anthropogenic stressors exceed the impact of the pesticides alone. These results are presented in Fig. S4 and discussed in the revised manuscript (Lines 349–360).

References

1. Fan, Y.; Chen, J.; Shirkey, G.; John, R.; Wu, S. R.; Park, H.; Shao, C., Applications of structural equation modeling (SEM) in ecological studies: an updated review. *Ecological Processes* 2016, 5, (1), 19.doi.org/10.1186/s13717-016-0063-3
2. Bliege Bird, R.; Coddling, B. F., Promise and peril of ecological and evolutionary modelling using cross-cultural datasets. *Nature Ecology & Evolution* 2022, 6, (1), 6-8.doi.org/10.1038/s41559-021-01579-w
3. Culina, A.; Baglioni, M.; Crowther, T. W.; Visser, M. E.; Woutersen-Windhouwer, S.; Manghi, P., Navigating the unfolding open data landscape in ecology and evolution. *Nature Ecology & Evolution* 2018, 2, (3), 420-426.doi.org/10.1038/s41559-017-0458-2

Comment 18: The interaction effects were measured using Hedges' *d*. Was there an attempt to apply nonlinear models (e.g., GAMs) to capture potential threshold effects in microbial responses?

Answer: We used generalized additive modeling (GAM) to assess the non-linear relationship between PBB diversity and multiple environmental and anthropogenic variables. The GAM results (are shown in Fig. S4) revealed clear threshold and unimodal effects—for example, pesticide usage exerted sharply negative impacts beyond moderate levels, and CO₂ emissions exhibited a nonlinear decline after a certain point. These findings underscore the importance of considering nonlinear ecological responses, and provide additional support for the notion that the synergistic effects of multiple anthropogenic pressures exceed the influence of pesticide effects alone. Relevant GAM results and interpretations have been added to the revised manuscript (lines 349–360).

Comment 19: The study reports a decline in carbon and nitrogen cycle genes. Were these shifts uniform across taxa, or did specific functional guilds (e.g., nitrogen-fixing bacteria) show greater sensitivity?

Answer: Our analysis focused on the functional gene changes within the significantly affected PBB, revealing that the decline in carbon and nitrogen cycle genes was not uniform across all taxa, but rather showed greater sensitivity in specific functional guilds, such as nitrogen-fixing bacteria (e.g., *Rhizobium* and *Bradyrhizobium*). These

taxa exhibited a more pronounced reduction in gene abundance than the other functional groups, likely because of their specialized metabolic roles and higher sensitivity to pesticide stress.

Comment 20: K01601_rbcL (carbon fixation gene) was significantly reduced. Was metatranscriptomic data available to confirm whether this leads to functional loss at the RNA level, or is this purely a metagenomic observation?

Answer: In our study, the reduction in K01601_rbcL abundance was based solely on metagenomic data, which reflect the presence and relative abundance of genes in the microbial community. However, we have no metatranscriptomic data to assess whether this reduction translates into functional loss at the RNA level (i.e., reduced gene expression). To illustrate this limitation, we provide a description and outlook in the Conclusion section (Lines 519–524).

Comment 21: The increase in sulfur-related genes is mentioned briefly. Could this indicate an adaptive response (e.g., sulfate reduction as an alternative electron acceptor)?

Answer: The increase in sulfur cycle genes could indicate an adaptive response of microbial communities to pesticide stress. Under conditions of pesticide-induced stress, certain microbes may shift their metabolic strategies to utilize sulfur compounds (e.g., sulfate reduction) as alternative electron acceptors, thereby maintaining energy production and survival in compromised environments.

In our study, we noted that an increase in sulfur cycle genes might reflect a shift in microbial strategies to cope with pesticide stress. This observation aligns with the findings of Ni et al.¹, who demonstrated that pesticide diversity significantly affects sulfur-cycling genes, such as those involved in polysulfide reduction (*cysJ*), dissimilatory sulfate reduction (*aprA*), and assimilatory sulfate reduction (*cysC*, *cysD*). These changes suggest that microbial communities compensate for disruptions in nitrogen and carbon cycling by enhancing sulfur-related metabolic pathways.

We have therefore revised the sentence “Under pesticide-induced stress, the increase in sulfur cycle genes may reflect a shift in microbial strategies to utilize sulfur compounds (e.g., sulfate reduction) as alternative electron acceptors, thereby maintaining energy production and survival. However, this adaptive response is unlikely to offset the broader ecosystem-level disruptions caused by the loss of essential carbon and nitrogen cycle functions.” in lines 419–423, making it more comprehensive.

Reference

1. Ni B, et al. Increasing pesticide diversity impairs soil microbial functions. *Proceedings of the National Academy of Sciences* **2025**, 122, e2419917122.

Comment 22: “17 pathways increased while 3 pathways decreased under high pesticide risk.” How do these shifts compare with previous studies on pesticide–microbiome interactions? Are there known resistance mechanisms upregulated in high–risk soils?

Answer: In our study, we observed that 17 functional pathways increased, whereas one decreased, under high pesticide risk. These shifts suggest that microbial communities adapt to pesticide stress by upregulating specific metabolic pathways, which potentially act as resistance mechanisms. While our study provides novel insights into these changes, direct comparisons with previous studies are challenging because of differences in experimental designs, pesticide types, and soil environments. Nevertheless, several past studies have reported similar trends in microbial functional responses to pesticide exposure, including the inhibition of key enzymatic activities (e.g., phosphatase) and alterations in microbial diversity under cumulative pesticide exposure¹, the upregulation of both carbohydrate metabolism and xenobiotic biodegradation pathways by rhizosphere microbiota to enhance stress resistance, the recruitment of organic-degrading bacteria and regulation of ABC transporters under co-exposure to multiple pesticides to mitigate pesticide uptake², and the reduction of specific microbial taxa alongside increased mineralization activity in response to pesticide mixtures, highlighting complex functional shifts³. Regarding resistance mechanisms, while our study did not explicitly identify specific upregulated resistance genes, previous research has highlighted potential mechanisms in high-risk soils, such as the upregulation of efflux pumps, enhanced biofilm formation, and increased horizontal gene transfer. Future studies should focus on elucidating the specific resistance mechanisms that are upregulated in high-risk soils, as defined by our framework.

References

1. Sim JXF, et al. Repeated applications of fipronil, propyzamide and flutriafol affect soil microbial functions and community composition: A laboratory-to-field assessment. *Chemosphere* 331, 138850 (2023)
2. Zhu Y, et al. Combined effects of azoxystrobin and oxytetracycline on rhizosphere microbiota of *Arabidopsis thaliana*. *Environment International* 186, 108655 (2024)
3. Mäder P, et al. Effects of MCPA and difenoconazole on glyphosate degradation and soil microorganisms. *Environmental Pollution* 362, 124926 (2024)

Comment 23: The study suggests that “growth factor biosynthesis genes decreased in pesticide-exposed PBB” Were specific genera responsible for these losses, or was the effect evenly distributed?

Answer: The observed decrease in growth factor biosynthesis genes was distributed across multiple PBB genera that were significantly affected by pesticide exposure rather than being confined to a few specific taxa. This indicated that the reduction in biosynthetic capacity was a widespread response among diverse PBB groups, suggesting a community-wide functional decline rather than a genus-specific effect.

Comment 24: “Amino acid biosynthesis genes declined.” Could this be due to pesticide toxicity directly affecting enzyme function, or does it suggest selective loss of amino-acid-producing microbes?

Answer: We investigated the functional potential of significantly altered PBB and

found a marked reduction in their amino acid biosynthesis genes. The observed decline is primarily attributed to pesticide-induced selective loss or suppression of amino acid-producing microbes, although we cannot exclude the potential direct toxic effects on enzymatic function, as our study focused specifically on functional gene loss in significantly altered PBBs and did not assess molecular-level impacts, such as post-translational modifications or enzyme activity.

The reduction in amino acid biosynthesis genes reflects a broader impact on microbial communities, where pesticide exposure may inhibit the survival or metabolic activity of the key PBB taxa responsible for amino acid production. This aligns with our observation that growth factor-related gene changes were distributed across significantly affected PBB, indicating a community-wide response to pesticide stress.

Comment 25: The study concludes that “targeted supplementation with amino acids and vitamins could restore PBB diversity.” Is there empirical evidence from past studies supporting this approach, or is this a speculative recommendation?

Answer: This recommendation is currently speculative and is based on the observed decline in amino acid and vitamin biosynthesis genes in pesticide-exposed soils.

Although there is limited direct empirical evidence from previous studies supporting this specific approach, the rationale is grounded in the well-established role of amino acids and vitamins as critical growth factors in microbial communities. For example, previous studies have shown that supplementing soils with specific nutrients (e.g., carbon sources and nitrogen compounds) can enhance microbial activity and diversity

under stress conditions¹. However, the targeted application of amino acids and vitamins to restore PBB diversity in pesticide-affected soils requires further experimental validation. In future work, we plan to conduct controlled experiments to test the efficacy of this approach and provide empirical evidence to support our hypotheses.

Reference

1. Lin J, Dai H, Yuan J, Tang C, Ma B, Xu J. Arsenic-induced enhancement of diazotrophic recruitment and nitrogen fixation in *Pteris vittata* rhizosphere. *Nature Communications* 15, 10003 (2024).

Comment 26: The conclusion should explicitly mention whether the authors anticipate long-term recovery of PBB if pesticide use is reduced, or if permanent microbial shifts are likely.

Answer: Thank you for your comment. In response to your comment, we have revised the Conclusion to explicitly address whether long-term recovery is anticipated or whether permanent microbial shifts are likely. Specifically, we have added a discussion on the potential for recovery and also the risk of irreversible changes in the microbial community composition due to prolonged pesticide exposure. These additions can be found in lines 529–536 in the revised manuscript.

Comment 27: The availability of R and Python scripts on GitHub is commendable, but were computational workflows benchmarked against alternative bioinformatics

pipelines (e.g., MG-RAST, QIIME2)?

Answer: Thank you for pointing this out. Although we did not directly benchmark our computational workflow against platforms such as MG-RAST or QIIME2, we have employed widely validated tools (e.g., Kraken2 for taxonomic profiling), each of which has been extensively used and tested in metagenomic studies. We have provided a supplementary explanation of this in the revised manuscript (Lines 164–170). Our pipeline was optimized for large-scale, high-throughput soil metagenomic datasets, and reproducibility was prioritized by providing all scripts and parameters on GitHub. In future work, we plan to perform formal benchmarking comparisons with other pipelines to further evaluate consistency and sensitivity.

Comment 28: The manuscript employs multiple statistical tests, but details on p-value corrections, assumptions of normality, and effect size interpretations are lacking.

Answer: We appreciate your suggestion to provide more details regarding the statistical methods used. In response to your comment, we made the following clarifications and revisions to the manuscript:

1. P-value Corrections: We acknowledge the importance of controlling for Type I errors when performing multiple comparisons. For the PERMANOVA and db-RDA models, we performed permutation tests ($n = 999$) to account for multiple testing, because the permutation approach inherently controls multiple comparisons.

Additionally, differential analysis of plant-beneficial bacteria (PBB) was performed

using STAMP software (v2.1.3). To compare the differences between treatment groups, we used Welch's *t*-test (two-sided), with p-values adjusted using the Benjamini-Hochberg false discovery rate (FDR) method in order to control for multiple comparisons. The significance threshold was set at $q < 0.05$, following FDR correction, ensuring that the false discovery rate was controlled across all comparisons.

2. Assumptions of normality: We understand that the assumption of normality is essential for many parametric tests. In the case of PERMANOVA and db-RDA, neither nonparametric test required normality assumptions. These tests are based on distance matrices (e.g., Bray-Curtis), which are suitable for analyzing compositional data without assuming a normal distribution.

3. Effect Size Interpretations: We agree that interpreting effect sizes is crucial for understanding the magnitude of the relationships between variables. In the manuscript, we have included the F-statistics and p-values from the PERMANOVA and db-RDA models as measures of the overall model significance. Additionally, we have included a discussion on the effect sizes of significant predictors, such as pesticide risk and spatial variables (PCNM), using their standardized coefficients, where applicable. Specifically, we added R values from the db-RDA (explained variation) to quantify the variation in community composition explained by each predictor. We have updated the manuscript to provide additional effect size interpretations in order to enhance our understanding of the strength of the observed relationships (Table S5).

Response to Reviewer #2 Comments:

Comment 1: In this study, Qiu and colleagues used metagenomic data from 1,919 global soil samples to identify 364 plant–beneficial bacteria. They found a higher PBB diversity in agricultural soils than in non–agricultural soils, but pesticide pollution reduced the abundance of PBB and their plant growth–promoting traits. They also found a loss of functional gene diversity in PBB with respect to important nutrient cycles, and amino acid and vitamin synthesis. They recommend the artificial application of specific amino acids and vitamins as a strategy to promote PBB in pesticide affected soils. While I like the overall idea, I do not think the manuscript is sufficiently novel for Nature Communications. I also have several concerns regarding the structure of the manuscript as well as the assumptions and methodology.

Answer: We sincerely thank the reviewer for the timely and thoughtful feedback. We understand the concerns regarding the perceived novelty of the manuscript. However, we would like to clarify that, to our knowledge, this is the first study to systematically assess the global patterns of plant-beneficial bacteria (PBB) diversity, function, and response to pesticide risk across 1,919 metagenomic soil samples obtained from diverse agricultural and non-agricultural sites worldwide.

While the plant growth-promoting role of microbes has been studied extensively at local and regional scales, there has been no comprehensive global assessment of how PBB responds to pesticide exposure and other anthropogenic pressures at a global level. The novelty of our study lies not only in the large, diverse dataset of global soil

samples, but also in our use of high-resolution ecological and genomic data to quantify the impacts of pesticide risk on microbial diversity, functional gene loss, and functional traits of PBB across various regions and ecosystems.

This study has provided an important and timely contribution in regard to the broader understanding of how anthropogenic factors, particularly pesticide pollution, shape PBB microbial communities. The insights gained from this study have significant implications for soil health management, sustainable agriculture, and ecosystem restoration globally. Specifically, our findings on the loss of functional genes in PBB due to pesticide exposure and the potential to mitigate these losses through targeted interventions (e.g., amino acid and vitamin supplementation) offer novel strategies for promoting microbial diversity and supporting agricultural sustainability.

In response to the reviewer's comments, we have carefully revised the manuscript to highlight the novel contributions and ensure greater clarity and coherence across sections. Specifically, we have:

1. Revised the Introduction to emphasize the knowledge gap in global-scale PBB research under pesticide stress and clearly articulate our hypotheses;
2. Improved the Methods section to more transparently describe our sample data strategy, bioinformatics pipeline, and statistical analyses;
3. Refined the Results to clearly differentiate findings on taxonomic diversity, functional traits, and environmental interactions;
4. Expanded the Discussion to more deeply explore the implications of functional gene loss, and the potential of targeted interventions (e.g., amino acid and vitamin

supplementation) to restore soil microbial health.

We are grateful for the reviewers' constructive suggestions, which have significantly improved the structure and clarity of the manuscript. We hope that the revised version now meets the standards of Nature Communications and that the novel insights it offers into pesticide–microbe interactions on a global scale will be of broad interest to the journal's readership.

Comment 2: My main concern is the lack of information regarding the identification of plant beneficial bacteria. I understand metaG data would reveal functional potential and not actual functions, however, it is not clear how they identified which ones are actually beneficial and at what level. Most soil bacteria are involved in biogeochemical (carbon, nitrogen cycling) processes and probably help with nutrient availability, but that doesn't make them plant beneficial. It is also expected that pesticides, which soil microorganisms could utilize as resources, would be correlated to C, N, and P cycling processes. The authors should consider beneficial functions such as N–fixing, P and K solubilizing, and biocontrol activities. Without knowing which bacteria are beneficial, subsequent analyses such as the World Development Index do not make sense

Answer: We thank the reviewer for raising this important concern regarding the identification of plant–beneficial bacteria (PBB) in our study. In order to address this, we have constructed a potential PBB database at the genus level by extracting bacteria from metagenomic samples following the approach of Li et al.¹ (Table S4). These

genus-level PBB were identified based on their documented roles in plantbeneficial activities, as supported by the study of Li et al. and other relevant literature.

In our subsequent analyses, we have classified these PBB into three functional categories based on their plant-beneficial properties: biocontrol, plant growth-promoting (PGP) activities (e.g., nitrogen fixation, phosphorus, and potassium solubilization), and stress resistance (e.g., drought and salinity tolerance). We then compared their abundance across different soil types (Fig. 1e). Furthermore, we investigated the functional potential of PBB by analyzing genes associated with plant-beneficial traits. We identified and compared three PGP-related genes (Fig. 4c). Although we acknowledge that most soil bacteria are involved in biogeochemical processes (e.g., carbon and nitrogen cycling), our focus on PBB was guided by their documented roles in direct plant-beneficial activities, as outlined above. This approach allowed us to distinguish PBB from general soil microbiota and assess their responses to pesticide stress in a targeted manner. We agree that future studies should further refine the identification of PBB by incorporating additional functional assays or metatranscriptomic data to validate their beneficial activities.

Reference

1. Li P, et al. Fossil-fuel-dependent scenarios could lead to a significant decline of global plant-beneficial bacteria abundance in soils by 2100. *Nature Food* 4, 996–1006 (2023).

Comment 3: It would be important to examine the effect on the overall soil bacterial

communities. This will reveal whether PGP were disproportionately affected in pesticide affected soils.

Answer: The broader effects of pesticides on overall soil bacterial communities have been comprehensively addressed in our previous study¹, which has now been cited in the revised manuscript (Lines 91–93). In this study, we analyzed 2,356 global soil metagenomes and developed a soil health–microbial index that integrates microbial diversity, nutrient cycling potential, metabolic capacity, primary productivity, and health risks to evaluate the responses of the entire soil microbiota to anthropogenic pressures. Our findings showed that pesticide contamination had the strongest negative impact on overall microbial health and that its effects were further exacerbated when combined with other anthropogenic and climatic factors. Notably, even under sustainable development scenarios, machine learning predictions suggest a continued decline in microbial health across approximately 26% of global farmlands. These results underscore the urgent need to address not only pesticide usage, but also the broader landscape of human-induced stressors to preserve soil microbial integrity on a global scale.

Building upon these findings, the current study specifically focused on the pesticide–associated responses of plant-beneficial bacteria (PBB) to provide a more detailed understanding of how key microbial taxa that directly contribute to soil functionality and plant health are affected by pesticide pressure. By narrowing our scope, we sought to uncover trait- and taxon-specific responses that were not the focus of the broader analysis, thereby complementing and extending our previous work.

Reference

1. Xu, N.; Chen, B.; Wang, Y.; Lei, C.; Zhang, Z.; Ye, Y.; Jin, M.; Zhang, Q.; Lu, T.; Dong, H.; Shou, J.; Penuelas, J.; Zhu, Y.-G.; Qian, H., Integrating Anthropogenic–Pesticide Interactions Into a Soil Health–Microbial Index for Sustainable Agriculture at Global Scale. *Global Change Biology* 2024, 30, (11).doi.org/10.1111/gcb.17596

Comment 4: Although it is a large dataset, the findings are largely correlative. Also, the dataset is just based on one time point, so the robustness of the reported correlations cannot be unsubstantiated.

Answer: We appreciate the comment. To assess the pesticide risk, we have utilized PEST–CHEMGRIDS¹, a comprehensive database that integrates temporal, geographical, and pesticide–specific data. This database estimates the global application rates of the 20 most-used pesticide active ingredients across six dominant crops and four aggregated crop classes at a 5 arc–minute resolution (~10 km at the equator). It relies on spatial statistical methods to reanalyze data from the USGS/PNSP, FAOSTAT, and other global datasets, incorporating soil properties, hydroclimatic variables, agricultural statistics, and socio–economic indices. PEST–CHEMGRIDS is based on data from 2000 to 2019 and provides projections from 2015 to 2025, ensuring that our dataset (2010–2023) falls within its coverage. Although we acknowledge that our findings are correlative in nature, the observed negative effects of pesticide risk on plant-beneficial bacteria align with previous

experimental studies^{2,3}, thus suggesting that the identified correlations may reflect the underlying causal mechanisms. Moreover, the use of the robust PEST-CHEMGRIDS database enhanced the reliability of our pesticide risk estimates. We also recognize that the lack of methodological detail was an oversight in the initial submission; this has now been addressed in the revised manuscript (Lines 192–193) with the addition of the following statement: “Pesticide risk calculations were based on PEST-CHEMGRIDSv1 data from 2000 to 2025.”

References

1. Maggi F, Tang FHM, la Cecilia D, McBratney A. PEST–CHEMGRIDS, global gridded maps of the top 20 crop-specific pesticide application rates from 2015 to 2025. *Scientific Data* 6, 170 (2019).
2. Walder, F.; Schmid, M. W.; Riedo, J.; Valzano-Held, A. Y.; Banerjee, S.; Buchi, L.; Bucheli, T. D.; van Der Heijden, M. G. A., Soil microbiome signatures are associated with pesticide residues in arable landscapes. *Soil Biology & Biochemistry* **2022**, 174.doi.org/10.1016/j.soilbio.2022.108830
3. Ke, M.; Xu, N.; Zhang, Z.; Qiu, D.; Kang, J.; Lu, T.; Wang, T.; Peijnenburg, W. J. G. M.; Sun, L.; Hu, B.; Qian, H., Development of a machine-learning model to identify the impacts of pesticides characteristics on soil microbial communities from high-throughput sequencing data. *Environmental Microbiology* **2022**, 24, (11), 5561-5573.doi.org/10.1111/1462-2920.16175

Comment 5: The manuscript could benefit from mechanistic discussions. There are

also no hypotheses or questions. Starting with the introduction, the authors should discuss how and why plant–growth promoting bacteria would be affected by pesticides. Do all pesticides affect in the same way, or would the effect vary with pesticide types? It would also be important to discuss the relative effects of active pesticide compounds vs residues. The discussion on the synergistic effects of pesticides and anthropogenic factors is strong but could explore potential molecular mechanisms behind these interactions. The authors should also expand on the long–term implications of functional loss in nutrient cycling genes and how these might cascade into broader ecosystem challenges.

Answer: We appreciate the reviewer’s thoughtful and constructive suggestions, which have helped us improve the depth and clarity of our manuscript significantly. In response, we have thoroughly revised the Introduction section to explicitly address the mechanisms by which plant growth–promoting bacteria (PBB) may be affected by pesticide exposure. Specifically, we discuss how different pesticide types, such as fungicides and herbicides, can disrupt microbial–plant symbioses, suppress beneficial taxa involved in plant and soil immunity, and select pesticide–resistant microbial specialists (Lines 55–74). We have also elaborated on how pesticide effects vary depending on their physicochemical properties (e.g., pKa, solubility, and molecular weight) and distinguished between the effects of active compounds and residues. The revised introduction includes new content addressing the ecological consequences of pesticide residues, which may persist longer in soils and influence microbial communities more subtly but extensively (Lines 68–74).

In response to the reviewer's concern about the lack of explicit hypotheses, we have clarified the main research objectives at the end of the Introduction (Lines 91–102), which reflect our underlying hypotheses: pesticide exposure alters the composition and function of PBB, these effects are modulated by other anthropogenic stressors, and targeted interventions may mitigate the observed impacts.

To further address the reviewer's request for mechanistic insights into the interaction between pesticides and anthropogenic pressures, we have added a discussion to the Results section (Lines 368–371) regarding the possible mechanisms driving the observed synergistic effects, such as increased pesticide leaching and bioavailability under high precipitation. Additionally, we have expanded the Introduction in order to discuss the long-term implications of microbial functional gene loss, especially those involved in nutrient cycling, and how such losses may cascade into broader disruptions in ecosystem services, including carbon sequestration and soil fertility (Lines 65–68).

Finally, we included new analyses using generalized additive models (GAMs) to explore the potential nonlinear and threshold responses of microbial diversity to anthropogenic drivers. The GAM results (detailed in Fig.S2 and described in Lines 349–360) revealed strong threshold effects and non-monotonic relationships, particularly for pesticide usage and precipitation, supporting the complexity of microbial responses to multi-stressor environments. These additions collectively strengthen the mechanistic and ecological interpretations of our findings in accordance with the reviewer's suggestions.

Comment 6: While the suggestion to supplement amino acids and vitamins is innovative, consider discussing potential barriers to implementation, such as cost, feasibility, and unintended ecological impacts.

Answer: We thank the reviewer for this valuable suggestion regarding the potential barriers to implementing amino acid and vitamin supplementation as a strategy for restoring PBB diversity and function. We agree that discussing the practical challenges and ecological implications of this approach is essential for translating our findings into practical solutions. In the revised manuscript, we have added a discussion on the potential barriers to implementation (lines 490–498), as follows:

1. **Cost:** The economic feasibility of large-scale amino acid and vitamin supplementation may be a significant challenge, particularly for small-scale farmers and those in low-resource agricultural settings.
2. **Feasibility:** The application methods, optimal dosages, and timing of supplementation need to be carefully optimized to ensure effectiveness without disrupting existing soil ecosystems.
3. **Unintended Ecological Impacts:** There Supplementing specific nutrients could alter microbial community composition in unintended ways, potentially favoring certain taxa over others and disrupting the ecological balance.

We also highlight the need for further research to address these challenges, including cost-benefit analyses, field trials to optimize application protocols, and ecological risk assessments to evaluate potential unintended consequences.

Comment 7: There is something strange about the PCoA analysis. The plot is showing a bizarre horseshoe pattern.

Answer: Thank you for your insightful comment regarding the PCoA plot. The observed horseshoe pattern in the PCoA analysis suggests a strong gradient effect, which can distort the representation of β -diversity. This effect often arises when linear ordination methods (such as PCoA) are applied to ecological data with a high beta diversity or compositional gradients.

In order to address this issue, we opted for nonmetric multidimensional scaling (NMDS) based on the Bray–Curtis distance for the following reasons:

1. Handling non-linearity:

NMDS is a rank-based, non-metric approach that is more robust to non-linear relationships and avoids artificial arching or horseshoe patterns that can arise in PCoA owing to strong compositional gradients.

2. Appropriate for Bray–Curtis dissimilarity:

The Bray–Curtis distance is commonly used in microbial community analyses because it effectively captures community compositional differences. Unlike Euclidean distances (which may exaggerate the influence of rare species' influence), Bray–Curtis focuses on relative abundance, making it more ecologically meaningful.

3. Better representation of ecological gradients:

NMDS optimizes ordination by preserving the rank-order relationships of distances rather than absolute distances, making it less susceptible to distortions caused by

strong gradients in the community composition.

4. Empirical improvement in visualization:

In our reanalysis, the NMDS plot provided a clearer, more interpretable structure of the microbial β -diversity patterns, avoiding the artificial distortion seen in PCoA.

Given these considerations, we revised the ordination analysis in our study to use BC-based NMDS (Fig. 2c), which provides a more accurate representation of community compositional differences. We appreciate your valuable feedback, which has helped us refine our approach.

Comment 8: The authors used confusing abbreviations— PGP, RPM, PBB. I think they should just call these PGP throughout, which is also more common in the literature.

Answer: We would like to clarify that the abbreviations PGP, PBB, and RPM refer to distinct concepts and are purposefully used to reflect their specific meanings.

1. PBB (Plant-Beneficial Bacteria) refers to a defined group of bacteria with functional roles in supporting plant growth, including nutrient cycling, stress tolerance, and disease suppression. This is the core subject of our study and the term encompasses both functional and taxonomic dimensions.
2. PGP (Plant Growth-Promoting) is a subset of traits or mechanisms (e.g., nitrogen fixation, phytohormone production, and phosphate solubilization) that characterize some PBB. We use this term specifically when referring to these biological activities or traits and not as a label for all taxa.

3. RPM (Reads Per Million) is a normalization metric used to represent the relative abundance of genes or taxa in metagenomic datasets. This is a standard unit used in sequencing-based studies and is unrelated to microbial functional classifications.

We agree that consistency in the terminology is important for clarity. In the revised manuscript, we have ensured that PGP is used consistently when referring to plant growth-promoting activities, whereas PBB will be retained to describe a broader category of plant-beneficial bacteria. We will also carefully review the manuscript to ensure that all abbreviations are clearly defined upon first use and are used consistently throughout the manuscript.

Comment 9: References are incomplete. All R packages and analysis tools must be cited with their original references.

Answer: Thank you for your comment. We have added original references for all of the R packages and tools cited in the manuscript, thus ensuring proper attribution to the developers and authors of these resources. These references have been added in the revised manuscript.

REVIEWERS' COMMENTS

Reviewer #1 (Remarks to the Author):

None

Response: We sincerely thank you for your time and effort in evaluating our revised manuscript. We appreciate the constructive feedback and suggestions provided during the previous review round, which have helped us to further improve the quality of the paper.